# DartQuant: Efficient Rotational Distribution Calibration for LLM Quantization

**Yuantian Shao**[1,2*] **Yuanteng Chen**[2,3,4*] **Peisong Wang**[2,3†] **Jianlin Yu**[5]
**Jing Lin**[5] **Yiwu Yao**[5] **Zhihui Wei**[1] **Jian Cheng**[2,3]
[1]Nanjing University of Science and Technology,
[2] C[2]DL, Institute of Automation, Chinese Academy of Sciences,
[3]School of Artificial Intelligence, University of Chinese Academy of Sciences,
[4]Zhongguancun Academy,
[5]Huawei Technologies Co., Ltd.

## Abstract

Quantization plays a crucial role in accelerating the inference of large-scale models, and rotational matrices have been shown to effectively improve quantization performance by smoothing outliers. However, end-to-end fine-tuning of rotational optimization algorithms incurs high computational costs and is prone to overfitting. To address this challenge, we propose an efficient distribution-aware rotational calibration method, DartQuant, which reduces the complexity of rotational optimization by constraining the distribution of the activations after rotation. This approach also effectively reduces reliance on task-specific losses, thereby mitigating the risk of overfitting. Additionally, we introduce the QR-Orth optimization scheme, which replaces expensive alternating optimization with a more efficient solution. In a variety of model quantization experiments, DartQuant demonstrates superior performance. Compared to existing methods, it achieves $47\times$ acceleration and $10\times$ memory savings for rotational optimization on a 70B model. Furthermore, it is the first to successfully complete rotational calibration for a 70B model on a single 3090 GPU, making quantization of large language models feasible in resource-constrained environments. Code is available at `https://github.com/CAS-CLab/DartQuant.git`.

## 1 Introduction

Large Language Models (LLMs) [1, 2, 3] have been a key breakthrough in natural language processing, demonstrating exceptional language understanding and generation capabilities through training on vast datasets with numerous parameters. These models perform exceptionally well on multiple tasks, including text generation, translation, and question-answering systems [4, 5]. However, the high computational and memory demands of LLM inference severely limit their deployment in resource-constrained environments [6, 7, 8, 9].

Current methods for reducing computational cost and improving the inference efficiency of deep learning models and LLMs include model pruning, knowledge distillation, parameter sharing, and quantization [10, 11, 12, 13, 14, 15]. Among these, post-training quantization (PTQ) stands out as a crucial technique to reduce computational costs due to its advantage of bypassing complex training processes, making it highly practical for real-world deployment [16, 17, 18, 19, 20].

---

[*]Equal contribution.
[†]Corresponding author.

39th Conference on Neural Information Processing Systems (NeurIPS 2025).

In LLM quantization, activations pose a greater challenge than weights due to the frequent presence of extreme outliers, which can significantly degrade model accuracy [21]. To address this issue, various outlier-handling techniques have been proposed. For example, high-bit protection mechanisms preserve the precision of outliers, while diagonal matrices help smooth extreme values in activations [22, 23]. Recent studies have shown that rotation matrices and affine transformations are highly effective in reducing outliers in activations, significantly improving quantization performance [24]. Rotation matrices are invertible, preserve vector norms, and can be seamlessly integrated into model architectures without introducing additional inference costs, making them a mainstream approach for quantization [25]. Although random Hadamard rotations can improve performance to some extent, they are not optimal. SpinQuant demonstrates that training rotation matrices further enhances quantization performance [26].

However, existing methods (e.g., SpinQuant [26], OSTQuant [27]) treat the rotation matrices as network parameters and fine-tune them end-to-end, which incurs substantial computational and memory costs associated with quantization. As shown in Figure 1, optimizing the rotation matrices for a 70B model requires hundreds of GiB of GPU memory and tens of gpu hours of computation, which conflicts with the fast deployment goals of PTQ algorithms. Moreover, end-to-end fine-tuning of rotation matrices presents unique challenges due to the complexity of optimizing on the rotation manifold. Specifically, rotation matrices must be carefully optimized to preserve orthogonality, which necessitates the use of specialized techniques such as Cayley or Riemannian SGD [28, 29]. These methods are computationally intensive and incur significant time overhead. Additionally, using small sample sizes for end-to-end fine-tuning poses a substantial risk of overfitting [27], which would worsen the optimization process.

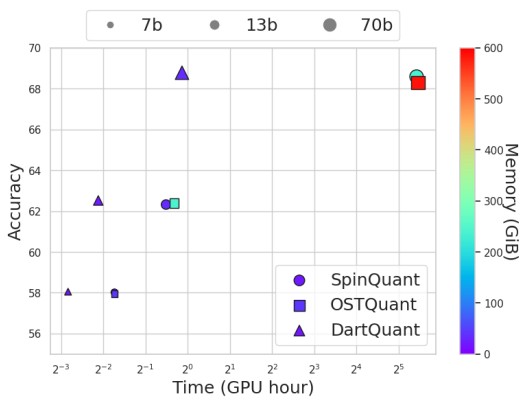

Figure 1: Comparison of computational costs across different rotation optimization methods.

To address these challenges, we propose DartQuant, a distribution-based rotation matrix calibration method that eliminates the need of end-to-end fine-tuning, significantly reducing the resource demands of rotation optimization while achieving higher accuracy. To mitigate overfitting, we redefine the rotation optimization problem from the perspective of distribution calibration, i.e., to transform the activations into the distribution most suitable for quantization. Based on the distribution transformation function that expands small value ranges and compresses large value ranges, we design the Whip loss. Unlike others that directly constrain outliers, Whip loss optimizes the activation distribution, making it more uniform and reducing the impact of outliers, thus lowering quantization errors. Finally, we introduce QR-Orth, an optimization method that applies orthogonal constraints using QR decomposition, avoiding complex projection calculations and significantly reducing the computational complexity of orthogonal optimization, thereby enhancing calibration efficiency.

Our contributions are summarized as follows:

- We introduce fast LLM Quantization with **rotational distribution calibration** framework, avoiding the excessive computational and memory costs of end-to-end fine-tuning paradigm. Based on this, the **Whip loss** is designed, which drives rotated activations toward a uniform distribution, effectively reducing quantization error and improving calibration efficiency.

- We present the **QR-Orth optimization** scheme, which ensures orthogonality of rotations via QR decomposition, eliminating the need for complex orthogonal optimizers. This reduces computational complexity and further enhances calibration efficiency.

- The proposed DartQaunt framework achieves superior quantization performance while significantly accelerating the rotation matrix calibration. For the 70B model, it **delivers a 47× speedup in terms of GPU hours and reduces memory usage by 10×** compared to existing methods. Notably, DartQuant enables rotation calibration of the **70B model on a single 3090 GPU in ∼3 hours**, greatly reducing calibration costs.

## 2 Related Work

### 2.1 Challenges in LLM Quantization

Large language models (LLMs) face challenges in quantization due to activation outliers, which take up most of the quantization range and reduce accuracy. To address this, researchers have proposed various strategies. Early methods used mixed precision, applying different precisions to weights and activations to reduce errors. However, mixed precision methods are complex and hinder inference speed and memory efficiency.

### 2.2 Outlier Handling through Scaling

To address outliers in activation quantization, scaling-based methods have been proposed. SmoothQuant [23] transfers outliers from activations to weights using scale invariance, reducing activation quantization errors. Outlier Suppression+ [21] addresses the asymmetric distribution of activations across channels by applying channel-wise scaling and shifting. OmniQuant [19] introduces learnable weight clipping and fine-tunes quantization errors using blockwise error minimization. Although scaling methods reduce outliers in activations, they often shift the quantization difficulty to weights. This does not fully solve the problem, especially in the presence of extreme outliers [30]. Efficiently handling outliers without complicating weight quantization remains a key challenge.

### 2.3 Outlier Handling through Rotation

Recent research shows that rotation matrices offer unique advantages in handling outliers in activation quantization. QuIP [31] first introduces incoherent processing to reduce the impact of outliers in both the weight and activation spaces. QuIP# [32] further improves speed by using randomized Hadamard transforms, which also have better theoretical properties. Building on this, QuaRot [25] combines the outlier suppression ability of rotation matrices with invariance transformations, applying it to large models like LLaMA, significantly improving PTQ performance. QuaRot also finds that Hadamard transforms outperform random orthogonal transformations in quantization. SpinQuant [26] extends the rotation matrix as a trainable parameter and employs the Cayley optimizer [28] for end-to-end fine-tuning. QServe [27] combines rotation and scaling techniques, using random Hadamard transforms and scaling methods to suppress outliers across different modules, enhancing performance under low-bit quantization. Further, OSTQuant [27] treats both rotation and scaling as trainable parameters and employs a KL-top loss for end-to-end fine-tuning, achieving better quantization accuracy.

Existing end-to-end fine-tuning methods for optimizing rotation matrices, while relatively simple to implement, typically incur significant optimization costs and are prone to overfitting. These methods also require high-quality calibration samples. Furthermore, the orthogonal optimizers used during optimization are computationally expensive, as they need to perform optimization on complex manifolds, adding additional challenges. In contrast, DartQuant significantly improves the efficiency of rotation matrix calibration. While maintaining comparable accuracy, it achieves a calibration speed that is more than $47\times$ faster than existing methods.

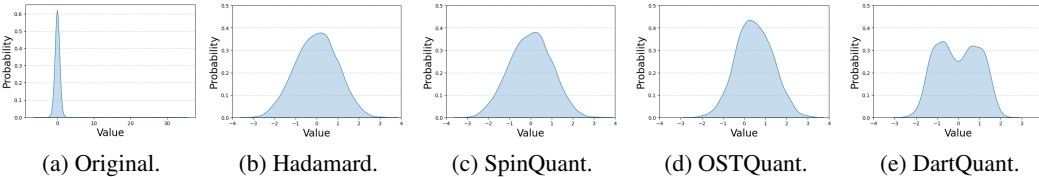

| (a) Original. | (b) Hadamard. | (c) SpinQuant. | (d) OSTQuant. | (e) DartQuant. |

Figure 2: Effects of different transformations on activation distribution.

## 3 Preliminaries and Difficulty

The Transformer architecture, commonly used in large language models (LLMs), consists of multi-head self-attention modules and feedforward network modules, both primarily composed of linear layers. Let the output of a linear layer be represented as $Y = XW^\top$, where $X \in \mathbb{R}^{T \times C_{\text{in}}}$ denotes the input activation, and $W \in \mathbb{R}^{C_{\text{out}} \times C_{\text{in}}}$ represents the weight matrix. Based on rotational invariance, we can insert an orthogonal transformation $R$ into the linear layer without altering the output, yielding

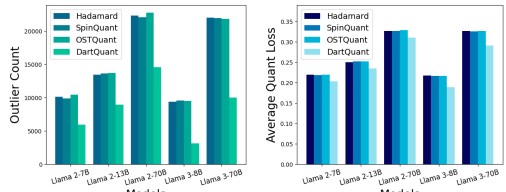

(a) The number of outliers.     (b) Quantization error.

Figure 3: Effects of different transformations on 1000 activations in layer 20 for various models. The rotation matrix optimized by DartQuant achieves the lowest number of outliers and the smallest quantization error.

Table 1: Impact of overfitting: Calibration on different data distribution on LLaMA models.

| Model | Datasets | WiKi | PTB | C4 |
|-------|----------|------|------|------|
| 2 7b | Baseline | 5.47 | 37.91 | 7.26 |
|       | WiKi | **5.94** | 45.13 | 8.13 |
|       | PTB | 6.02 | **38.24** | 8.13 |
|       | C4 | 6.05 | 44.99 | **8.02** |
| 2 13b | Baseline | 4.88 | 50.94 | 6.73 |
|       | WiKi | **5.21** | 58.39 | 7.40 |
|       | PTB | 5.33 | **49.14** | 7.41 |
|       | C4 | 5.33 | 60.59 | **7.32** |

$Y = (XR)(R^\top W^\top)$, where $R \in \mathbb{R}^{C_{\text{in}} \times C_{\text{in}}}$ is an orthogonal matrix satisfying $RR^\top = I$. By combining $R$ with the previous weight matrix and $R^\top$ with the current layer's weight matrix, we can rotate the activation vector without introducing any additional computational cost.

Similarly, under the condition that the model's output remains unchanged, we can insert four orthogonal matrices $R_1, R_2, R_3, R_4$ within the Transformer block, **as outlined in [26] and Appendix A**. Specifically, by multiplying $R_1$ on the right side of $W_q, W_k, W_v, W_{\text{up}}, W_{\text{gate}}$, and multiplying $R_1^\top$ on the left side of $W_{\text{out}}$ and $W_{\text{down}}$, an equivalent transformation is achieved. Similarly, $R_2$ can be inserted between $W_v$ and $W_{\text{out}}$. $R_3$ can be inserted between the rotated encodings of $Q$ and $K$. $R_4$ can be inserted before $W_{\text{down}}$. Finally, $W_{\text{embedding}}$ is multiplied on the left by $R_1^T$, and $W_{\text{lm\_head}}$ is multiplied on the right by $R_1$, completing all the equivalent transformations. This process is referred to as the "Computational Invariance" [33].

QuaRot [25] demonstrates that using random Hadamard rotation can achieve good results; however, this is not optimal, as shown in Figure 2. Methods like SpinQuant [26] and OSTQuant [27] treat the rotation matrix $R$ as learnable network parameters and fine-tune them in an end-to-end manner with pseudo-quantizers inserted, resulting in better quantization performance. Although end-to-end fine-tuning is simple to implement, the complex computational process faces resource challenges and optimization difficulties. In Figure 3a, we present the number of outliers among 1000 activations in the 20th layer of various models after different transformations. Figure 3b shows the average quantization error of these samples (statistics for activations from other layers are in Appendix F). It is evident that the transformations in end-to-end fine-tuning do not significantly reduce the number of outliers in the activations, nor do they notably lower the quantization error, highlighting the limitations of end-to-end fine-tuning.

Moreover, end-to-end fine-tuning based on calibration sets not only consumes considerable computational resources but also tends to lead to overfitting on the calibration set [34, 27]. As shown in Table 1, fine-tuning methods exhibit a significant performance improvement on the corresponding test sets, with the improvement being particularly pronounced on the PTB dataset. A possible explanation is that the limited and relatively simple calibration data often fail to fully cover the parameter space of large models, causing the model to overfit to a narrow feature distribution, thereby limiting its generalization and emergent capabilities [27].

In addition, optimizing the rotation matrix requires the use of orthogonal optimizers to ensure the matrix's orthogonality. These optimizers are based on Riemannian optimization on the Grassmanian or Stiefel manifold, involving complex projection computations [28, 29]. As a result, their computational time is approximately twice that of standard optimizers. This, in turn, further increases the cost of rotation optimization and slows down the optimization process.

## 4 Method

In this section, we provide a detailed description of the proposed DartQuant method. DartQuant is comprised of three key components: the rotational distribution calibration, the Whip loss function, and QR-Orth optimization. Each of these components addresses the primary challenges outlined earlier. Figure 4 illustrates the overall framework, highlighting the flow and interactions between these components.

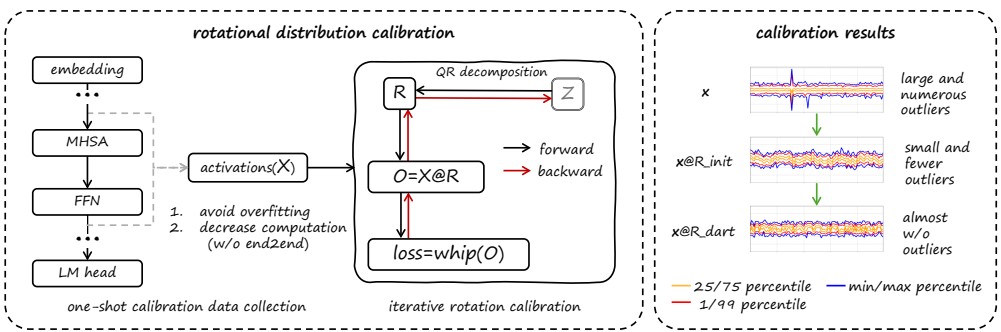

Figure 4: Left: The DartQuant implementation process, with $Z$ representing the latent parameters in QR-orth and $R$ as the applied rotation matrix. Right: The change in rotation matrix before and after calibration.

## 4.1 Rotational Distribution Calibration

End-to-end fine-tuning typically requires more data, and the optimization of rotation matrices depends on the task-specific loss, which significantly increases the risk of overfitting. For LLMs, end-to-end fine-tuning also entails substantial computational and memory overheads. To address these challenges, we redefine the rotation optimization problem and propose a rotational distribution calibration.

Specifically, we revisit the optimization objective of the rotation matrix from the perspective of feature distribution transformation. We redefine the problem as finding a rotation matrix that transforms the activations into the distribution most suitable for quantization. This approach reduces the reliance on task-specific loss during calibration, thereby mitigating the risk of overfitting.

Previous studies have shown that outliers are the primary cause of activation quantization loss. Therefore, we constrain the activation distribution after rotation by minimizing the number of outliers in the transformed activations, i.e.

$$\min_R \sum_{i=1}^{c_{in}} \mathbb{I}(|(Rx)_i| > \tau) \tag{1}$$

The function $\mathbb{I}(\cdot)$ represents the indicator function, and $\tau$ is the threshold used to identify outliers. Although this problem cannot be directly solved using standard stochastic gradient descent, we can resort to approximation methods for calibration. In statistics, variance is commonly used to measure the dispersion of data; however, using variance as an optimization objective is not ideal. Due to the symmetric distribution of activations [34], the variance of activations typically corresponds to a constant multiple of the activation vector's norm square. Furthermore, the norm-invariance property of the rotation matrix introduces significant challenges when directly optimizing using variance (as shown in Figure 7a). In addition to variance, kurtosis is frequently used to measure the heaviness of the distribution's tails, making it a suitable alternative objective. However, since the rotated activations are already close to a Gaussian distribution with relatively few outliers, optimizing with kurtosis is slow (as shown in Figure 7a). Therefore, there is an urgent need for a better optimization objective to constrain the activation distribution.

## 4.2 Activation Uniformity via Whip Loss

To better reduce outliers, we propose a new optimization objective that constrains the activation distribution to approach a uniform distribution, thereby effectively reducing the number of outliers in the rotated activations.

As shown in Figure 2a and Appendix G, the activation tokens exhibit a distribution near Laplace. Assuming that the activation tokens follow a Laplace distribution with mean $\mu = 0$ and scale parameter $b$, the probability density function (PDF) is given by:

$$f(x) = \frac{1}{2b} \exp(-\frac{|x|}{b}). \tag{2}$$

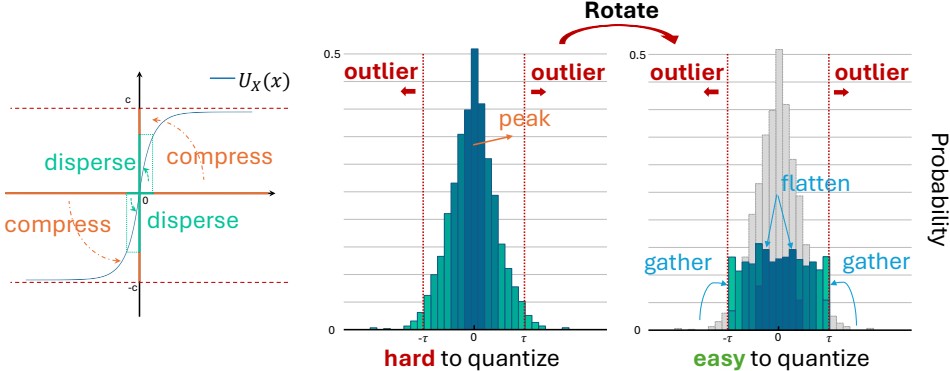

Figure 5: Intuition behind the distribution transformation: $U_X(x)$ transforms the Laplace distribution into a uniform distribution by flattening the peak and aggregating the outliers.

In statistics, cumulative distribution functions (CDFs) are often used to transform one distribution into another [35]. To convert $x \sim Laplace(0, b)$ to a uniform distribution over the interval $[-\tau, \tau]$, the transformation function is the following:

$$
\begin{aligned}
U_X(x) &= 2\tau \left[ \int_{-\infty}^{x} \frac{1}{2b} \exp(-\frac{|x|}{b})\, dt - \frac{1}{2} \right] \\
&= \begin{cases} \tau[\exp(\frac{x}{b}) - 1], & x \leq 0, \\ \tau[1 - \exp(-\frac{x}{b})], & x > 0. \end{cases}
\end{aligned}
\tag{3}
$$

As shown in Figure 5, the left side presents the function graph of $U_X(x)$, where the intervals near the origin are expanded, while those further from the center are compressed. The right side visually illustrates the impact of this transformation on the distribution. $U_X(x)$ spreads values originally concentrated around the center over a wider range, thus smoothing the peak of the distribution. Meanwhile, outliers farther from the center are gathered together, shrinking the overall distribution range, ultimately resulting in a uniform distribution within the interval $[-\tau, \tau]$.

Inspired by the mechanism of $U_X(x)$, we propose the Whip loss function:

$$
Whip = \sum_{i=1}^{c_{in}} \exp(-|x_i|).
\tag{4}
$$

Here, $\boldsymbol{x} = [x_1, x_2, \ldots, x_{C_{in}}] \in \mathbb{R}^{C_{in}}$ denotes the activation vector. Clearly, the Whip function is continuously differentiable, and has larger gradients near zero. When used as a loss function, smaller values in the rotated activation vector are pushed away from zero. In other words, the Whip function encourages the rotation to smooth the sharp central peak of the Laplace distribution, producing a more uniform distribution. As the magnitudes of several small-value channels in the activation vector increase, the outliers are suppressed due to the norm-invariance constraint. This results in an "aggregation" effect in the activation values. As a result, the overall activation distribution tends to converge toward a uniform distribution within a smaller interval, thereby effectively reducing the quantization error.

## 4.3 Enforcing Orthogonality with QR-Orth

To satisfy the orthogonality constraint, the rotation matrix must be optimized on the Grassmannian or Stiefel manifold, which necessitates the use of specialized optimizers, such as the Cayley SGD [28] used in SpinQuant [26]. Unlike gradients in Euclidean space, gradients on manifolds require complex projection operations, resulting in significantly higher computational costs. To avoid the computational complexity of orthogonal optimizers, we propose the QR-Orth optimization method.

Specifically, we can obtain an orthogonal matrix $R \in \mathbb{R}^{n \times n}$ and an upper triangular matrix $U \in \mathbb{R}^{n \times n}$ by performing a QR decomposition on any matrix $Z \in \mathbb{R}^{n \times n}$. Based on this relationship, we design

the rotational distribution calibration method with QR-Orth optimizer shown in Algorithm 1. We use the orthogonal matrix $R$, obtained from the QR decomposition, as the rotation matrix for the actual computation. The latent parameter $Z$ is treated as a optimization parameter and is discarded after calibration. By optimizing the latent matrix $Z$, we indirectly optimize the rotation matrix $R$. In this way, we can use any optimizer to optimize the rotation matrix.

When the matrix size becomes large, the Cayley optimizer introduces a computational overhead of approximately $6n^3$ compared to standard optimizers. In contrast, QR-Orth only incurs the cost of the QR decomposition, with a computational complexity on the order of $\frac{4}{3}n^3$ (see Appendix B for a detailed complexity derivation). Although QR decomposition usually requires iterative calculations, the significant reduction in overall computational load has brought about a 1.4x acceleration effect. In practice, QR-Orth can easily integrate with various optimizers such as SGD or Adam, making it highly adaptable. This flexibility makes QR-Orth a promising solution for optimizing orthogonal matrices.

---

**Algorithm 1** Rotational Distribution Calibration with QR-Orth Optimizer

1: **Input:** LLM model $LLM$, calibration sequence $S$, initial latent parameter $Z_0 \in \mathbb{R}^{n \times n}$, max iterations $T$, learning rate $\eta$.
2: **Output:** Rotational matrix $R \in \mathbb{R}^{n \times n}$.
3: $X \leftarrow LLM(S)$
4: $X \leftarrow token\_sampling(X)$
5: $Z \leftarrow Z_0$
6: **for** $k = 0$ to $T$ **do**
7: $\quad R \leftarrow qr\_decomposition(Z)$
8: $\quad O \leftarrow X@R$
9: $\quad \mathcal{L} \leftarrow Whip(O)$
10: $\quad Z \leftarrow Z - \eta \frac{\partial \mathcal{L}}{\partial Z}$
11: **end for**

---

## 5 Experiment

**Model and Dataset.** We evaluate our method on the Llama series models, including Llama-2 (7B/13B/70B) [1] and Llama-3 (8B/70B). Moreover, we also provide results on two popular MoE models: Mixtral-8x7B [36] and Deepseek-MoE [37]. We report perplexity (PPL) scores on the WikiText2 [38], C4 [39], and PTB [40]. Additionally, we assess model performance on nine zero-shot evaluation tasks, including LAMBADA [41], HellaSwag [42], PIQA [43], WinoGrande [44], OpenBookQA [45], SIQA [46], MMLU [47], ARC-E, and ARC-C [48].

**Baselines and Implementation Details.** In addition to the basic RTN method, we compare our approach with several other methods, including SmoothQuant [23], GPTQ [49], OmniQuant [19], and current state-of-the-art methods such as Quarot [25], SpinQuant [26] and OSTQuant [27] for weight and activation quantization.

In the main results, we apply GPTQ to reconstruct the weights. To do so, we use 128 samples from WikiText2, with a sequence length of 2048 tokens, as the calibration set for GPTQ, following the standard GPTQ setup. All activations are quantized using per-token asymmetric quantization. We optimize all orthogonal matrices using SGD combined with QR-Orth. During the orthogonal matrix calibration phase, we use 128 samples from WikiText2, each with a token length of 2048.

### 5.1 Main Results

Table 2 evaluates six models across four common bit-width settings, offering practical guidance for selecting appropriate rotation schemes. DartQuant utilizes learned rotation matrices $R_1$ and $R_2$ , which can be fused into the model weights during inference, eliminating any additional computational overhead. In contrast, online Hadamard rotations ($R_3$ and $R_4$) leverage fast Hadamard kernels for efficient inference computation [32]. As shown in Table 2, when both weights and activations are quantized to 8 bits, the performance differences among methods are minimal. However, when weights are quantized to 4 bits and activations to 8 bits, methods like SmoothQuant and OmniQuant experience significant performance degradation. This is primarily due to SmoothQuant's design, which complicates weight quantization and increases quantization errors, leading to a substantial drop in performance. In contrast, other methods generally maintain the model accuracy.

Although rotation transformation methods improve quantization performance when activations are quantized to 8 bits, the additional computational cost associated with $R_3$ and $R_4$ makes this approach less efficient. When activations are quantized to 4 bits, omitting the rotation matrix results in a significant performance drop. Furthermore, DartQuant, SpinQuant, and OSTQuant, which optimize

Table 2: Comparison of the average Perplexity Scores across three datasets and the average accuracy on nine Zero-shot Common Sense Reasoning tasks. The results for all comparison methods were obtained using their publicly available codebases. Full results can be found in the Appendix C.

| Bits (W-A-KV) | Method | Llama-2 7B | | Llama-2 13B | | Llama-2 70B | | Llama-3 8B | | Llama-3 70B | |
|---|---|---|---|---|---|---|---|---|---|---|---|
| | | PPL ↓ | 0-shot[9] ↑ | PPL ↓ | 0-shot[9] ↑ | PPL ↓ | 0-shot[9] ↑ | PPL ↓ | 0-shot[9] ↑ | PPL ↓ | 0-shot[9] ↑ |
| 16-16-16 | FloatingPoint | 16.88 | 61.16 | 20.85 | 64.28 | 11.09 | 69.53 | 8.92 | 66.04 | 6.19 | 72.70 |
| 4-8-16 | RTN | 18.15 | 59.37 | 21.77 | 62.49 | 12.53 | 67.83 | 10.35 | 62.97 | 12.38 | 67.18 |
| | SmoothQuant | 332.17 | 30.97 | 1510.66 | 29.89 | 180.96 | 38.36 | 112.46 | 31.94 | 544.68 | 33.95 |
| | GPTQ | 6977.62 | 60.03 | 20.76 | 63.70 | 11.90 | 69.03 | 10.27 | 64.79 | 6.89 | 69.44 |
| | OmniQuant | 426.53 | 59.15 | **20.74** | 62.95 | 14.06 | 67.18 | 10.48 | 62.72 | 14.95 | 59.94 |
| | QuaRot | 18.41 | 59.92 | 22.02 | 63.50 | **11.15** | 69.09 | 9.59 | 64.92 | 6.92 | 70.75 |
| | SpinQuant | 17.85 | 60.10 | 21.15 | 63.53 | 11.29 | **69.57** | **9.48** | 65.01 | **6.63** | 71.76 |
| | DartQuant | **17.69** | **60.17** | 20.93 | **63.77** | 11.18 | 69.30 | 9.49 | **65.58** | 6.66 | **71.82** |
| 4-4-16 | RTN | 668.50 | 31.39 | 2523.84 | 29.61 | 63311.10 | 29.12 | 200.56 | 30.54 | 17390.85 | 31.01 |
| | SmoothQuant | 3278.95 | 29.48 | 4366.47 | 29.05 | 1636.53 | 29.40 | 2216.28 | 29.55 | 6242.62 | 29.30 |
| | GPTQ | 1529.13 | 31.05 | 1554.72 | 29.85 | 68684.35 | 29.23 | 270.49 | 33.47 | 14201.63 | 32.53 |
| | OmniQuant | 202.95 | 40.18 | 107.01 | 42.98 | 109.27 | 41.08 | 186.02 | 31.25 | 380.94 | 28.37 |
| | QuaRot | 20.63 | 57.90 | 24.11 | 61.81 | **11.35** | 67.92 | 11.74 | 58.20 | 10.73 | 62.28 |
| | SpinQuant | 19.90 | 57.85 | 22.88 | 62.32 | 11.70 | 68.59 | 10.67 | 62.29 | 9.61 | 66.06 |
| | OSTQuant | 19.24 | 57.94 | **22.33** | 62.38 | 11.98 | 68.29 | 10.66 | 62.18 | **7.67** | 67.94 |
| | DartQuant | **18.53** | **58.05** | 22.44 | **62.64** | 11.51 | **69.02** | 10.58 | 62.80 | 7.99 | **69.39** |
| 4-4-4 | RTN | 853.68 | 30.29 | 2535.13 | 29.53 | 63772.42 | 29.15 | 353.44 | 30.54 | 17803.40 | 30.41 |
| | GPTQ | 1813.34 | 29.97 | 1929.93 | 29.53 | 78362.25 | 29.15 | 496.93 | 31.60 | 17361.71 | 32.98 |
| | QuaRot | 27.01 | 57.03 | 24.98 | 59.87 | **11.49** | 67.41 | 12.29 | 57.32 | 11.38 | 61.50 |
| | SpinQuant | 25.12 | 57.55 | 23.37 | 61.60 | 11.76 | 68.05 | 10.99 | 61.35 | 10.17 | 64.76 |
| | OSTQuant | 19.74 | 57.88 | 22.83 | 62.31 | 11.67 | 68.11 | **10.66** | 61.57 | **7.76** | 67.84 |
| | DartQuant | **19.14** | **57.96** | 22.64 | **62.46** | 11.55 | **68.22** | 10.78 | 62.38 | 8.13 | **69.05** |

rotation matrices, clearly demonstrate the necessity of rotation matrix optimization, outperforming QuaRot. Additionally, while SpinQuant and OSTQuant perform well in reducing perplexity, their performance in zero-shot tasks is poor, which highlights the potential overfitting risks associated with end-to-end fine-tuning methods. In contrast, DartQuant, with its novel calibration strategy, generates rotation matrices that effectively compress the activation distribution range, resulting in outstanding performance in 0-shot tasks. Notably, we achieve a performance loss of only 0.5% on Llama 2-70b under w4a4kv16 setting. For Llama 3-70b which is more difficult to quantize [50], we manage to limit the average performance loss to 3.31%, outperforming SpinQuant and OSTQuant by 3.33% and 1.45% respectively.

**Besides dense LLMs, we also extend our research to recent popular MoE-LLMs, and further experimental results can be found in the Appendix H.**

Table 3 presents a comparison of the optimization time and memory consumption of SpinQuant, OSTQuant, and DartQuant on an A800 GPU server. DartQuant simplifies the calibration framework, leading to significant reductions in resource overhead across various models. In particular, for the 70B model, DartQuant completes the calibration in 30 minutes using a single GPU, achieving a **speedup of 47× in training and 10× in memory savings** compared to Spin-

Table 3: Comparison of Rotation Matrix Optimization Cost.

| Cost | Method | 7B | 13B | 70B |
|---|---|---|---|---|
| Time (GPU hour) | SpinQuant | 0.30 | 0.70 | 42.90 |
| | OSTQuant | 0.30 | 0.80 | 44.00 |
| | DartQuant | 0.14 | 0.23 | 0.91 |
| | DartQuant$_{3090}$ | 0.43 | 0.70 | 2.90 |
| Memory (GiB) | SpinQuant | 19.98 | 33.73 | 238.89 |
| | OSTQuant | 42.25 | 239.16 | 583.86 |
| | DartQuant | 17.41 | 21.40 | 23.47 |
| | DartQuant$_{3090}$ | 17.41 | 21.40 | 23.47 |

Quant and OSTQuant. Moreover, DartQuant is the first to optimize the rotation matrix of the 70B model on **a single 3090 GPU**, with a calibration time of ∼**3 hours**. This development substantially reduces the cost of rotation matrix optimization and enhances its practical value.

## 5.2 Ablation Studies

We compared the effectiveness of four optimization objectives: quantization loss, variance, kurtosis, and the Whip function. As shown in Figure 7a, the change in activation quantization loss over iteration steps is presented for each optimization objective. It is clearly observed that when using quantization loss, variance, or kurtosis as the optimization objective, the activation quantization loss shows minimal variation. However, when the Whip function is used as the optimization objective, the quantization loss curve decreases significantly within fewer iterations and converges rapidly.

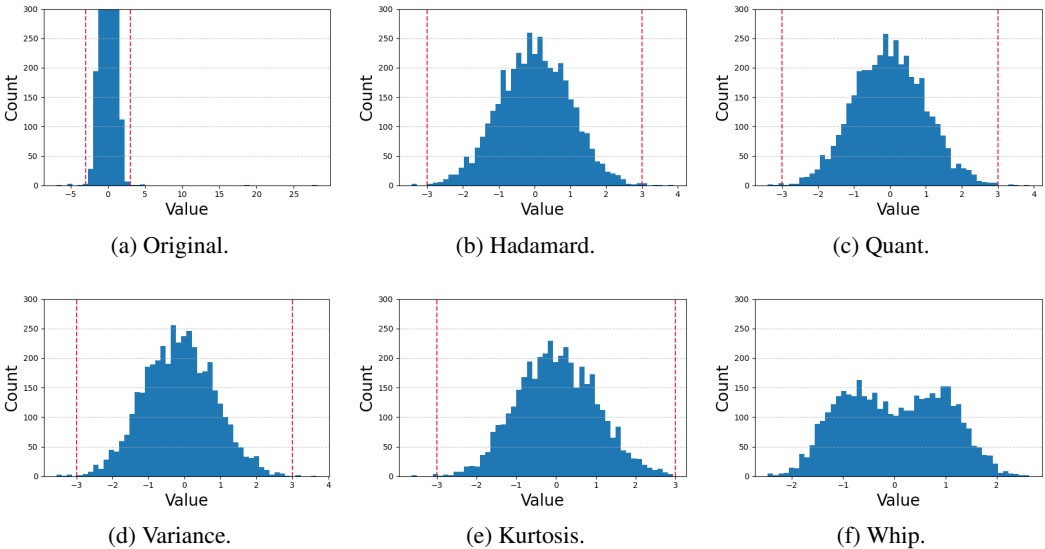

Figure 6: Histograms of Activation Distributions After Rotation by Different Rotation Matrices. The region outside the red dashed line represents the outliers.

### 5.2.1 Optimization objectives

By comparing the effects of different optimization objectives on the activation distribution (as shown in Figure 6), we can clearly observe the substantial changes induced by the Whip function. Figure 6a shows the histogram of the original, unrotated activation distribution. From the range on the x-axis, it is evident that the original distribution contains significant outliers. Figure 6b displays the histogram of the activation distribution after a random Hadamard matrix rotation. After the Hadamard rotation, the activation range is notably compressed, although some outliers remain untreated. The rotation matrices trained with quantization loss and variance as optimization objectives show little improvement, resulting in an activation distribution nearly identical to that obtained by random Hadamard rotation. Although kurtosis optimization slightly improves the distribution, its effect is limited. In contrast, the histogram after Whip optimization shows a significant improvement (as shown in Figure 6f): this method not only effectively addresses the outlier problem but also disperses the activation points, initially concentrated around zero, across other regions. The resulting distribution is the closest to a uniform distribution. This outcome aligns closely with our design goals and further validates the effectiveness of our approach. More ablation studies on different optimization objectives under zero-shot tasks and perplexity metrics are provided in Appendix I.

### 5.2.2 Optimizer Comparison

Figure 7b presents the comparison of the convergence curves between the Cayley optimization and our proposed QR-Orth optimization, both using Whip loss under identical settings. It is evident that QR-Orth demonstrates faster convergence and lower final loss, regardless of whether combined with SGD or Adam. As shown in Table 4, QR-Orth achieves a $1.4\times$ speedup over Cayley optimization for the same number of iterations. Due to its faster convergence, QR SGD achieves the same result as Cayley SGD after 100 steps in just 6 steps, yielding an overall acceleration factor of $41\times$. This significantly improves the efficiency of orthogonal optimization.

Table 4: Comparison of Time Taken for 100 Iterations Across Different Orthogonal Optimization Schemes.

| Method | Cayley | QR-Orth | Speed up |
|---|---|---|---|
| SGD | 8.2h | 5.7h | 1.44x |
| Adam | 8.1h | 5.7h | 1.42x |

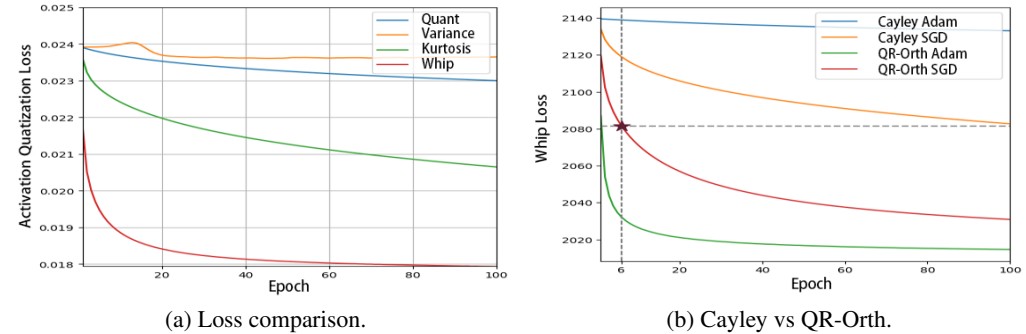

| (a) Loss comparison. | (b) Cayley vs QR-Orth. |

Figure 7: Comparison of activation quantization loss and convergence performance using different optimization methods.

## 5.3 Results on Different Datasets

To investigate the sensitivity of DartQuant to different training datasets, we sample training data from three datasets: WIKITEXT2, PTB, and C4. These datasets are used to optimize R1 and R2 separately, and the impact of the dataset on the performance of the quantized LLM is compared. As shown in Table 5, the results of all three experiments are largely consistent. This demonstrates that DartQuant is robust to calibration datasets and does not negatively affect the generalization ability of the LLM. Further analysis on the impact of sample size on performance is provided in Appendix D.

Table 5: Comparison of LLM Performance with Different Calibration Datasets Using DartQuant.

| Model | Dataset | WikiText2 | PTB | C4 | Avg |
|---|---|---|---|---|---|
| 2 7b | Baseline | 5.47 | 37.91 | 7.26 | 16.88 |
| | WikiText2 | 5.92 | 42.63 | 7.99 | 18.85 |
| | PTB | 5.91 | 42.78 | 8.01 | 18.90 |
| | C4 | 5.92 | 42.99 | 8.00 | 18.97 |
| 2 13b | Baseline | 4.88 | 50.94 | 6.73 | 20.85 |
| | WikiText2 | 5.25 | 58.29 | 7.3 | 23.61 |
| | PTB | 5.24 | 58.46 | 7.33 | 23.68 |
| | C4 | 5.28 | 58.18 | 7.31 | 23.59 |

## 6 Conclusions

In this paper, we introduce DartQuant, an innovative method for LLM quantization that efficiently handles activation outliers. By constraining the activation distribution, DartQuant simplifies rotation matrix calibration and avoids overfitting risks in end-to-end fine-tuning. The QR-Orth optimization method eliminates the need for complex orthogonal optimization, further speeding up the process. DartQuant achieves state-of-the-art results in 4-bit quantization while significantly reducing costs. Notably, it successfully quantizes a 70B model on a single RTX 3090, advancing LLM deployment in resource-constrained environments.

## 7 Acknowledgments

This work was supported in part by the National Key R&D Program of China (No. 2025ZD0122000), the Science and Technology Major Special Program of Jiangsu (No.BG2024028), Beijing Natural Science Foundation (L244046), the Jiangsu Key Research and Development Plan (No. BE2023016), and the CCF-Baidu Open Fund.

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

# A Computational Invariance in Transformers

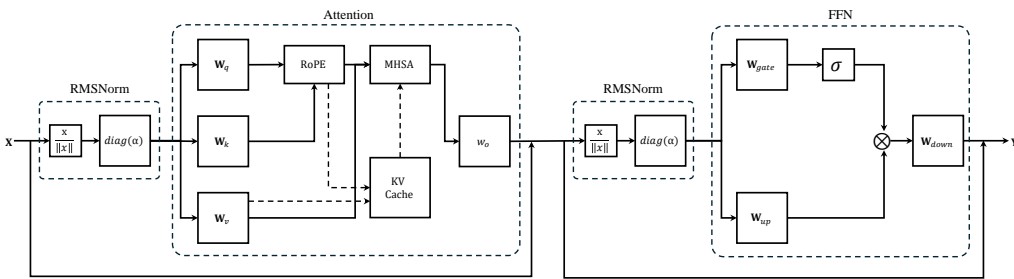

Figure 8: Flowchart of the transformer block used in most language models, including pre-RMSNorm, multi-head self-attention (MHSA), and the gated feedforward network (FFN). The solid arrows represent the data flow during training, pre-filling, and inference for each token. In RMSNorm, the input signal is normalized by its norm and rescaled by the parameter $\sigma$. In MHSA, the RoPE module computes the relative position embeddings, and the dashed arrows indicate access to the KV-Cache during generation. In the FFN, the activation function $\sigma$ is applied to the gated signal, and the two signals are combined element-wise.

The weights and activations between blocks in the transformer can be transformed using orthogonal matrices without altering the model's output. Specifically, if a rotation transformation $R_1$ is applied to the input activations, the effect can be offset by left-multiplying the weight matrices on the left side of the transformer block (such as $W_q, W_k, W_v, W_{up}, W_{gate}$ in Figure 8) by the orthogonal matrix $R_1^\top$. To ensure the correctness of the residual computation, $R_1$ is right-multiplied with the output matrices (such as $W_o$ and $W_{down}$ in Figure 8). Although RMSNorm is applied between two blocks, this transformation remains valid as long as there is no rescaling within RMSNorm (in practice, any rescaling is typically absorbed into the adjacent weight matrices). Theoretically, this is because RMSNorm only normalizes the activations, and applying a rotation transformation does not alter the norm of the activations. As a result, the commutative property $RMSNorm(XR_1) = RMSNorm(X)R_1$ can be established [33].

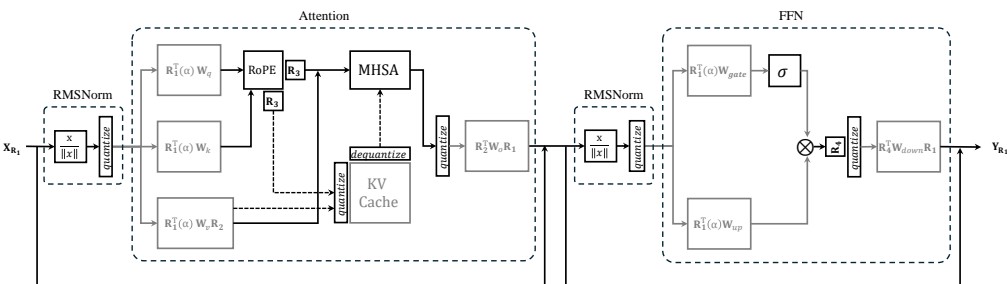

Figure 9: Transformer applied in DartQuant. The RMSNorm scaling factor ($\sigma$) has been absorbed into the weight matrices. The black section represents the flow in FP16 format, while the gray section indicates the flow in INT4 format, and the dashed line shows the flow in and out of the KV buffer. The hidden state $X$ has been rotated by $R_1$, which is offset by $R_1^\top$ and absorbed into the weight matrices $W_q, W_k, W_v, W_{up}, W_{gate}$. $R_1$ is also incorporated into $W_o$ and $W_{down}$ to ensure correct residual calculation. $R_2$ in $W_v$ cancels with $R_2^\top$ in $W_o$. $R_3$ and $R_4$ are random Hadamard matrices computed online: $R_3$ cancels out during the attention computation, and $R_4$ cancels with $R_3^\top$ absorbed in $W_{down}$. All weights are stored in INT4 format, and all activations prior to the weights are also quantized to INT4. The result of the matrix multiplication between INT4 weights and INT4 activations on TensorCore is INT32, which is immediately converted (and scaled) to FP16.

In addition to $R_1$ , the Transformer block can also incorporate $R_2$, $R_3$, and $R_4$ as shown in Figure 9. $R_2$ operates within the multi-head attention mechanism by being applied to each attention head. It is absorbed into $W_v$ to rotate the input of $W_o$. To counterbalance the rotation effect introduced by $W_v$, the transpose of $R_2$, denoted $R_2^\top$, is left-multiplied to $W_o$.

$R_3$ is absorbed into the KV cache to alleviate the quantization loss within the KV cache. Due to the presence of RoPE, directly incorporating $R_3$ into the weight matrices is challenging, so $R_3$ is designed as an online Hadamard transform.

Finally, $R_4$ is a rotation matrix used to smooth the input to the down-projection. Given the gating mechanism, its rotational component cannot be fused into $W_{up}$ or $W_{gate}$, thus it is also designed as an online Hadamard transform. However, the inverse transformation can be combined with $W_{down}$, avoiding the computational overhead of the inverse operation.

**It is important to emphasize that the inference framework of DartQuant (Figure 9) is identical to the framework proposed by SpinQuant [26], resulting in the same level of inference acceleration.**

## B  Comparison of Calculation Amount

### B.1  QR Decomposition

The Householder QR decomposition for an $n \times n$ matrix $A$ can be expressed as follows:

---
**Algorithm 2** Householder QR Decomposition

---
1: **Input:** Matrix $A \in \mathbb{R}^{n \times n}$
2: **Output:** Orthogonal matrix $Q \in \mathbb{R}^{n \times n}$, Upper triangular matrix $R \in \mathbb{R}^{n \times n}$
3: Initialize $R = A$
4: **for** $k = 1$ to $n - 1$ **do**
5:   Set the vector $x = R_{k:n,k}$
6:   Compute the Householder vector $v = x + \text{sign}(x_1)\|x\|_2 e_1$
7:   Normalize $v$: $v = \frac{v}{\|v\|_2}$
8:   Update $R$: $R_{k:n,k:n} = R_{k:n,k:n} - 2vv^T R_{k:n,k:n}$
9:   Update the orthogonal matrix $Q$: $Q_{k:n,k:n} = Q_{k:n,k:n} - 2vv^T Q_{k:n,k:n}$
10: **end for**
11: $R_{n,n}$ is the upper triangular matrix, $Q$ is the orthogonal matrix.

---

Let us analyze the computational complexity of each step in the above pseudocode:

- **Step 6&7: Computing the Householder vector** $v$:
    - The operation involves computing the norm $\|x\|_2$, which takes $(n - k + 1)$.
    - Computing $v$ involves adding two vectors and normalizing, which is also $(n - k + 1)$.
- **Step 8: Updating** $R$:
    - The update of $R$ requires multiplying a vector $v$ by a matrix slice $R_{k:n,k:n}$, which takes $(n - k + 1)^2$.
- **Step 9: Updating** $Q$:
    - Updating $Q$ involves a similar computation to updating $R$, so it also takes $(n - k + 1)^2$.

We now aggregate the computational complexity of all steps across the iterations. The overall computational complexity is given by:

$$
\begin{aligned}
\text{Total complexity} &\approx \sum_{k=1}^{n-1} 2[(n - k + 1)^2 + (n - k + 1)] \\
&= \frac{2n(n + 1)(2n + 1)}{6} + n(n + 1) - 4 \\
&\approx \frac{4}{3}n^3
\end{aligned}
\tag{5}
$$

## B.2 Cayley SGD

Algorithm 3 provides the detailed computational process of the Cayley SGD. Compared to standard SGD, it introduces additional computations in steps 5 through 12. These computations are primarily composed of matrix multiplications.

---

**Algorithm 3** Cayley SGD with Momentum

---

1: learning rate $l$, momentum coefficient $\beta$, $\epsilon = 10^{-8}$, $q = 0.5$, $s = 2$
2: Initialize $X_1$ as an orthonormal matrix; and $M_1 = 0$
3: **for** $k = 0$ to $T$ **do**
4:     $M_{k+1} \leftarrow \beta M_k - \mathcal{G}(X_k)$
5:     $\hat{W}_k \leftarrow M_{k+1} X_k^\top - \frac{1}{2} X_k (X_k^\top M_{k+1} X_k^\top)$
6:     $W_k \leftarrow \hat{W}_k - W_k^\top$
7:     $M_{k+1} \leftarrow W_k X_k$
8:     $\alpha \leftarrow \min\{l, 2q/(\|W_k\| + \epsilon)\}$
9:     Initialize $Y^0 \leftarrow X + \alpha M_{k+1}$
10:    **for** $i = 1$ to $s$ **do**
11:        $Y^i \leftarrow X_k + \frac{\alpha}{2} W_k (X_k + Y^{i-1})$
12:    **end for**
13:    Update $X_{k+1} \leftarrow Y^s$
14: **end for**

---

- **Step 5: Computing the auxiliary matrix $\hat{W}_k$**

  - The first part involves computing $M_{k+1} X_k^\top$, which requires a matrix multiplication with complexity $n^3$.

  - The second part involves the term $X_k (X_k^\top M_{k+1} X_k^\top)$. To optimize this computation, the matrix multiplication $X_k^\top M_{k+1} X_k^\top$ requires $n^3$, and multiplying this with $X_k$ results in a total complexity of $2n^3$ due to the nested operations.

- **Step 7: Momentum projection update**

  - The update $M_{k+1} = W_k X_k$ is a matrix multiplication, which takes $n^3$.

- **Step 9 to 11: Iterative Cayley transform**

  - Each iteration involves matrix addition and matrix multiplication, which has a complexity of $n^3$.

  - Since there are $s$ iterations, the total complexity is $2n^3$.

In conclusion, Cayley SGD incurs approximately an additional $6n^3$ in computational complexity compared to standard SGD.

## C   Complete Results of Main Result Table

In Tables tables 6 to 15, we present the full results of Table 2. We report perplexity (PPL) scores on the WikiText2 [38], C4 [39], and PTB [40]. Additionally, we assess model performance on nine zero-shot evaluation tasks: LAMBADA [41], HellaSwag [42], PIQA [43], WinoGrande [44], OpenBookQA [45], SIQA [46], MMLU [47], ARC-E, and ARC-C [48]. We compare our approach with several other methods, including SmoothQuant [23], GPTQ [49], OmniQuant [19], and current state-of-the-art methods such as Quarot [25] and SpinQuant [26] for weight and activation quantization.

Table 6: Comprehensive comparison of the average accuracy of LLaMA-2 7B on nine Zero-Shot Commonsense Reasoning tasks.

| Bits (W-A-KV) | Method | WG | SIQA | PIQA | OBQA | LAMB | HS | ARC-E | ARC-C | MMLU | Avg |
|---|---|---|---|---|---|---|---|---|---|---|---|
| 16-16-16 | Full Precision | 69.06 | 46.16 | 79.05 | 44.2 | 73.9 | 76.02 | 74.54 | 46.33 | 41.21 | 61.16 |
| 4-8-16 | RTN | 68.51 | 44.88 | 78.24 | 42.20 | 70.50 | 74.07 | 73.06 | 44.62 | 38.24 | 59.37 |
| | SmoothQuant | 50.75 | 34.14 | 55.22 | 26.40 | 0.82 | 32.09 | 31.61 | 24.91 | 22.80 | 30.97 |
| | GPTQ | 69.53 | 45.55 | 78.78 | 42.80 | 72.29 | 74.60 | 73.40 | 44.11 | 39.21 | 60.03 |
| | OmniQuant | 69.22 | 44.73 | 77.91 | 42.40 | 71.84 | 73.88 | 72.60 | 43.26 | 36.52 | 59.15 |
| | QuaRot | 68.27 | 44.88 | 77.86 | 43.80 | 73.96 | 75.23 | 72.90 | 43.6 | 38.77 | 59.92 |
| | SpinQuant | 67.96 | 44.83 | 78.78 | 43.60 | 72.95 | 74.80 | 73.61 | 44.80 | 39.55 | 60.10 |
| | DartQuant | 68.75 | 45.55 | 78.78 | 42.60 | 73.63 | 75.15 | 73.65 | 44.62 | 38.81 | 60.17 |
| 4-4-16 | RTN | 50.59 | 35.82 | 53.65 | 29.80 | 6.21 | 29.73 | 30.89 | 22.44 | 23.39 | 31.39 |
| | SmoothQuant | 49.25 | 35.36 | 49.89 | 26.80 | 0.00 | 25.60 | 27.23 | 26.62 | 24.61 | 29.48 |
| | GPTQ | 50.28 | 35.47 | 53.05 | 27.40 | 3.59 | 29.17 | 32.03 | 25.26 | 23.22 | 31.05 |
| | OmniQuant | 51.85 | 37.67 | 63.55 | 32.60 | 28.22 | 49.43 | 46.04 | 27.47 | 24.81 | 40.18 |
| | QuaRot | 66.06 | 43.45 | 76.44 | 42.60 | 71.26 | 73.57 | 69.87 | 43.09 | 34.80 | 57.90 |
| | SpinQuant | 66.06 | 43.71 | 76.61 | 39.80 | 72.17 | 73.70 | 70.96 | 42.83 | 34.79 | 57.85 |
| | OSTQuant | 66.69 | 45.60 | 76.71 | 43.60 | 70.62 | 73.64 | 69.36 | 40.96 | 34.32 | 57.94 |
| | DartQuant | 67.17 | 44.93 | 76.93 | 39.00 | 71.65 | 73.76 | 70.96 | 42.41 | 35.66 | 58.05 |
| 4-4-4 | RTN | 49.88 | 34.75 | 52.56 | 26.20 | 3.07 | 28.44 | 28.91 | 25.34 | 23.46 | 30.29 |
| | GPTQ | 50.91 | 33.98 | 52.39 | 24.60 | 1.92 | 28.21 | 30.51 | 23.63 | 23.61 | 29.97 |
| | QuaRot | 65.27 | 43.65 | 77.04 | 40.60 | 70.25 | 72.81 | 68.43 | 41.64 | 33.56 | 57.03 |
| | SpinQuant | 64.17 | 44.73 | 76.44 | 40.60 | 71.16 | 73.40 | 70.50 | 42.32 | 34.67 | 57.55 |
| | OSTQuant | 66.93 | 43.76 | 77.37 | 40.40 | 71.24 | 73.50 | 69.95 | 42.58 | 35.22 | 57.88 |
| | DartQuant | 66.69 | 44.68 | 77.48 | 38.60 | 70.95 | 73.81 | 69.87 | 44.20 | 35.32 | 57.96 |

Table 7: Comprehensive comparison of Perplexity scores for LLaMA-2 7B across three datasets.

| Bits (W-A-KV) | Method | Wiki | PTB | C4 | Avg |
|---|---|---|---|---|---|
| 16-16-16 | Full Precision | 5.47 | 37.91 | 7.26 | 16.88 |
| 4-8-16 | RTN | 5.91 | 40.64 | 7.89 | 18.15 |
| | SmoothQuant | 250.33 | 500.08 | 246.09 | 332.17 |
| | GPTQ | 7.69 | 20917.14 | 8.04 | 6977.62 |
| | OmniQuant | 5.75 | 1266.1 | 7.75 | 426.53 |
| | QuaRot | 5.63 | 42.00 | 7.60 | 18.41 |
| | SpinQuant | 5.61 | 40.41 | 7.53 | 17.85 |
| | DartQuant | 5.60 | 39.95 | 7.52 | 17.69 |
| 4-4-16 | RTN | 487.93 | 758.10 | 759.48 | 668.50 |
| | SmoothQuant | 2999.11 | 2426.98 | 4410.75 | 3278.95 |
| | GPTQ | 995.88 | 2497.44 | 1094.07 | 1529.13 |
| | OmniQuant | 17.06 | 566.12 | 25.68 | 202.95 |
| | QuaRot | 6.02 | 47.64 | 8.24 | 20.63 |
| | SpinQuant | 5.89 | 45.66 | 8.14 | 19.90 |
| | OSTQuant | 5.89 | 43.76 | 8.06 | 19.24 |
| | DartQuant | 5.88 | 41.72 | 7.99 | 18.53 |
| 4-4-4 | RTN | 731.05 | 819.7 | 1010.3 | 853.68 |
| | GPTQ | 1905.56 | 2254.27 | 1280.19 | 1813.34 |
| | QuaRot | 6.17 | 66.41 | 8.46 | 27.01 |
| | SpinQuant | 5.99 | 61.03 | 8.34 | 25.12 |
| | OSTQuant | 5.94 | 45.16 | 8.13 | 19.74 |
| | DartQuant | 5.93 | 43.41 | 8.08 | 19.14 |

Table 8: Comprehensive comparison of the average accuracy of LLaMA-2 13B on nine Zero-Shot Commonsense Reasoning tasks.

| Bits (W-A-KV) | Method | WG | SIQA | PIQA | OBQA | LAMB | HS | ARC-E | ARC-C | MMLU | Avg |
|---|---|---|---|---|---|---|---|---|---|---|---|
| 16-16-16 | Full Precision | 72.14 | 47.39 | 80.52 | 45.20 | 76.73 | 79.38 | 77.44 | 49.15 | 50.53 | 64.28 |
| 4-8-16 | RTN | 71.67 | 46.01 | 79.22 | 43.60 | 75.06 | 75.51 | 74.96 | 48.38 | 48.00 | 62.49 |
| | SmoothQuant | 51.38 | 34.54 | 53.65 | 26.20 | 0.35 | 27.21 | 28.20 | 24.57 | 22.95 | 29.89 |
| | GPTQ | 72.30 | 46.57 | 80.09 | 45.60 | 76.25 | 78.13 | 76.73 | 48.29 | 49.32 | 63.70 |
| | OmniQuant | 70.88 | 46.11 | 79.76 | 44.60 | 47.56 | 75.04 | 77.70 | 76.60 | 48.29 | 62.95 |
| | QuaRot | 70.72 | 46.26 | 79.49 | 44.80 | 76.62 | 78.52 | 76.39 | 49.15 | 49.59 | 63.50 |
| | SpinQuant | 71.11 | 46.21 | 79.76 | 44.80 | 76.85 | 78.30 | 76.56 | 48.98 | 49.21 | 63.53 |
| | DartQuant | 72.06 | 46.88 | 79.98 | 44.40 | 76.81 | 78.73 | 76.85 | 48.98 | 49.26 | 63.77 |
| 4-4-16 | RTN | 47.83 | 33.93 | 52.45 | 27.40 | 0.72 | 26.88 | 29.29 | 24.23 | 23.72 | 29.61 |
| | SmoothQuant | 49.41 | 34.08 | 50.65 | 24.20 | 0.00 | 25.77 | 26.09 | 27.39 | 23.89 | 29.05 |
| | GPTQ | 51.30 | 34.60 | 52.56 | 23.60 | 1.26 | 27.29 | 30.68 | 23.38 | 23.97 | 29.85 |
| | OmniQuant | 53.67 | 39.51 | 67.68 | 33.00 | 30.39 | 54.90 | 53.58 | 29.69 | 24.42 | 42.98 |
| | QuaRot | 69.69 | 45.14 | 78.73 | 43.80 | 74.29 | 76.24 | 74.75 | 47.18 | 46.43 | 61.81 |
| | SpinQuant | 70.24 | 45.75 | 78.78 | 44.20 | 74.36 | 77.50 | 75.88, | 46.84 | 47.36 | 62.32 |
| | OSTQuant | 69.46 | 46.37 | 78.89 | 44.00 | 74.62 | 77.13 | 75.21 | 47.95 | 47.79 | 62.38 |
| | DartQuant | 71.11 | 46.16 | 79.27 | 44.20 | 75.18 | 78.04 | 75.38 | 47.61 | 46.80 | 62.64 |
| 4-4-4 | RTN | 50.12 | 33.98 | 50.33 | 25.60 | 0.60 | 26.78 | 28.96 | 25.68 | 23.73 | 29.53 |
| | GPTQ | 50.04 | 34.19 | 50.33 | 25.00 | 0.58 | 27.87 | 29.34 | 24.66 | 23.73 | 29.53 |
| | QuaRot | 69.30 | 44.27 | 78.24 | 42.80 | 65.44 | 75.75 | 72.64 | 45.90 | 44.52 | 59.87 |
| | SpinQuant | 68.35 | 45.65 | 77.75 | 43.60 | 73.28 | 77.10 | 75.00 | 47.61 | 46.08 | 61.60 |
| | OSTQuant | 70.40 | 46.37 | 79.16 | 45.00 | 75.20 | 76.74 | 73.53 | 47.18 | 47.19 | 62.31 |
| | DartQuant | 71.11 | 45.29 | 79.27 | 43.80 | 74.93 | 77.57 | 76.01 | 47.53 | 46.59 | 62.46 |

Table 9: Comprehensive comparison of Perplexity scores for LLaMA-2 13B across three datasets.

| Bits (W-A-KV) | Method | Wiki | PTB | C4 | Avg |
|---|---|---|---|---|---|
| 16-16-16 | Full Precision | 4.88 | 50.94 | 6.73 | 20.85 |
| 4-8-16 | RTN | 5.23 | 52.95 | 7.12 | 21.77 |
| | SmoothQuant | 1884.63 | 920.66 | 1726.69 | 1510.66 |
| | GPTQ | 5.00 | 50.39 | 6.90 | 20.76 |
| | OmniQuant | 5.07 | 50.13 | 7.02 | 20.74 |
| | QuaRot | 5.02 | 54.08 | 6.96 | 22.02 |
| | SpinQuant | 4.99 | 51.56 | 6.91 | 21.15 |
| | DartQuant | 4.98 | 50.89 | 6.91 | 20.93 |
| 4-4-16 | RTN | 1595.89 | 3016.34 | 2959.29 | 2523.84 |
| | SmoothQuant | 4811.32 | 4299.06 | 3989.04 | 4366.47 |
| | GPTQ | 914.38 | 1856.09 | 1893.68 | 1554.72 |
| | OmniQuant | 15.39 | 282.92 | 22.73 | 107.01 |
| | QuaRot | 5.35 | 59.52 | 7.46 | 24.11 |
| | SpinQuant | 5.19 | 56.07 | 7.37 | 22.88 |
| | OSTQuant | 5.21 | 54.46 | 7.33 | 22.33 |
| | DartQuant | 5.22 | 54.82 | 7.28 | 22.44 |
| 4-4-4 | RTN | 1793.38 | 2845.81 | 2966.21 | 2535.13 |
| | GPTQ | 1367.96 | 2758.47 | 1663.36 | 1929.93 |
| | QuaRot | 5.51 | 61.78 | 7.65 | 24.98 |
| | SpinQuant | 5.30 | 57.3 | 7.51 | 23.37 |
| | OSTQuant | 5.25 | 55.87 | 7.37 | 22.83 |
| | DartQuant | 5.26 | 55.32 | 7.34 | 22.64 |

Table 10: Comprehensive comparison of the average accuracy of LLaMA-2 70B on nine Zero-Shot Commonsense Reasoning tasks.

| Bits (W-A-KV) | Method | WG | SIQA | PIQA | OBQA | LAMB | HS | ARC-E | ARC-C | MMLU | Avg |
|---|---|---|---|---|---|---|---|---|---|---|---|
| 16-16-16 | Full Precision | 77.98 | 49.13 | 82.75 | 48.8 | 79.58 | 83.8 | 81.02 | 57.51 | 65.2 | 69.53 |
| 4-8-16 | RTN | 76.01 | 47.19 | 82.37 | 48.00 | 77.59 | 80.64 | 80.77 | 56.74 | 61.14 | 67.83 |
| | SmoothQuant | 53.35 | 34.34 | 69.80 | 33.60 | 9.70 | 41.60 | 50.34 | 28.41 | 24.10 | 38.36 |
| | GPTQ | 77.74 | 47.70 | 82.75 | 49.00 | 79.57 | 82.83 | 81.27 | 56.74 | 63.68 | 69.03 |
| | OmniQuant | 73.95 | 46.72 | 81.45 | 48.40 | 75.06 | 82.64 | 80.64 | 57.08 | 58.66 | 67.18 |
| | QuaRot | 77.27 | 48.31 | 82.75 | 48.00 | 80.01 | 83.21 | 80.64 | 57.00 | 64.61 | 69.09 |
| | SpinQuant | 78.22 | 48.93 | 82.48 | 48.60 | 80.26 | 83.90 | 81.52 | 57.76 | 64.44 | 69.57 |
| | DartQuant | 78.14 | 48.82 | 82.81 | 47.80 | 80.13 | 83.4 | 80.6 | 57.51 | 64.49 | 69.30 |
| 4-4-16 | RTN | 50.28 | 34.19 | 50.05 | 26.20 | 0.00 | 25.67 | 26.56 | 26.11 | 23.02 | 29.12 |
| | SmoothQuant | 48.93 | 33.83 | 50.65 | 26.60 | 0.52 | 27.21 | 27.57 | 24.66 | 24.63 | 29.40 |
| | GPTQ | 49.96 | 34.80 | 50.82 | 25.00 | 0.12 | 25.72 | 26.85 | 25.26 | 24.58 | 29.23 |
| | OmniQuant | 52.88 | 37.62 | 63.71 | 33.20 | 25.01 | 51.89 | 50.76 | 30.89 | 23.76 | 41.08 |
| | QuaRot | 75.93 | 47.80 | 82.15 | 46.60 | 78.98 | 81.98 | 80.64 | 56.14 | 61.07 | 67.92 |
| | SpinQuant | 76.56 | 48.93 | 82.15 | 46.60 | 79.53 | 83.00 | 80.98 | 56.66 | 62.89 | 68.59 |
| | OSTQuant | 76.24 | 47.95 | 82.32 | 47.20 | 79.20 | 82.52 | 80.09 | 56.31 | 62.78 | 68.29 |
| | DartQuant | 77.58 | 48.52 | 82.70 | 48.20 | 79.99 | 82.62 | 81.93 | 57.00 | 62.60 | 69.02 |
| 4-4-4 | RTN | 49.96 | 34.19 | 50.38 | 23.60 | 0.00 | 25.96 | 26.35 | 28.92 | 23.02 | 29.15 |
| | GPTQ | 50.59 | 34.03 | 49.46 | 25.60 | 0.16 | 25.86 | 26.39 | 26.28 | 23.96 | 29.15 |
| | QuaRot | 76.48 | 46.26 | 81.28 | 46.00 | 79.00 | 81.82 | 79.46 | 55.63 | 60.78 | 67.41 |
| | SpinQuant | 75.93 | 47.85 | 81.50 | 47.40 | 79.16 | 82.90 | 79.91 | 55.20 | 62.59 | 68.05 |
| | OSTQuant | 76.58 | 48.21 | 82.32 | 46.60 | 79.58 | 81.70 | 80.51 | 56.23 | 61.28 | 68.11 |
| | DartQuant | 76.40 | 48.93 | 82.81 | 47.00 | 79.76 | 82.56 | 78.58 | 55.38 | 62.55 | 68.22 |

Table 11: Comprehensive comparison of Perplexity scores for LLaMA-2 70B across three datasets.

| Bits (W-A-KV) | Method | Wiki | PTB | C4 | Avg |
|---|---|---|---|---|---|
| 16-16-16 | Full Precision | 3.32 | 24.25 | 5.71 | 11.09 |
| 4-8-16 | RTN | 3.75 | 27.72 | 6.11 | 12.53 |
| | SmoothQuant | 95.26 | 318.96 | 128.66 | 180.96 |
| | GPTQ | 3.50 | 26.33 | 5.87 | 11.90 |
| | OmniQuant | 4.01 | 31.78 | 6.40 | 14.06 |
| | QuaRot | 3.42 | 24.23 | 5.80 | 11.15 |
| | SpinQuant | 3.40 | 24.68 | 5.79 | 11.29 |
| | DartQuant | 3.41 | 24.34 | 5.78 | 11.18 |
| 4-4-16 | RTN | 65680.98 | 60069.34 | 64182.98 | 63311.10 |
| | SmoothQuant | 1868.60 | 1906.48 | 1134.50 | 1636.53 |
| | GPTQ | 69089.76 | 68684.35 | 79127.77 | 72300.63 |
| | OmniQuant | 27.29 | 265.00 | 35.53 | 109.27 |
| | QuaRot | 3.78 | 24.16 | 6.10 | 11.35 |
| | SpinQuant | 3.67 | 25.38 | 6.04 | 11.70 |
| | OSTQuant | 3.65 | 26.28 | 6.01 | 11.98 |
| | DartQuant | 3.64 | 24.90 | 5.99 | 11.51 |
| 4-4-4 | RTN | 66102.31 | 60357.61 | 64857.35 | 63772.42 |
| | GPTQ | 73603.62 | 75839.09 | 85644.05 | 78362.25 |
| | QuaRot | 3.81 | 24.53 | 6.14 | 11.49 |
| | SpinQuant | 3.69 | 25.52 | 6.07 | 11.76 |
| | OSTQuant | 3.67 | 25.3 | 6.03 | 11.67 |
| | DartQuant | 3.67 | 24.98 | 6.01 | 11.55 |

Table 12: Comprehensive comparison of the average accuracy of LLaMA-3 8B on nine Zero-Shot Commonsense Reasoning tasks.

| Bits (W-A-KV) | Method | WG | SIQA | PIQA | OBQA | LAMB | HS | ARC-E | ARC-C | MMLU | Avg |
|---|---|---|---|---|---|---|---|---|---|---|---|
| 16-16-16 | Full Precision | 73.32 | 47.19 | 80.96 | 45.00 | 75.63 | 79.14 | 77.82 | 53.24 | 62.06 | 66.04 |
| 4-8-16 | RTN | 70.88 | 45.04 | 78.29 | 46.60 | 70.17 | 76.86 | 73.95 | 49.23 | 55.75 | 62.97 |
| | SmoothQuant | 51.70 | 34.39 | 55.71 | 24.20 | 8.97 | 33.30 | 33.84 | 22.10 | 23.27 | 31.94 |
| | GPTQ | 72.45 | 46.57 | 79.38 | 46.40 | 73.47 | 77.60 | 76.43 | 51.11 | 59.70 | 64.79 |
| | OmniQuant | 70.64 | 44.83 | 79.16 | 43.40 | 68.93 | 76.75 | 73.65 | 48.81 | 58.30 | 62.72 |
| | QuaRot | 74.11 | 46.83 | 79.49 | 43.80 | 74.11 | 77.07 | 77.40 | 51.79 | 59.64 | 64.92 |
| | SpinQuant | 74.19 | 45.91 | 79.92 | 44.80 | 75.74 | 77.10 | 77.10 | 51.88 | 60.74 | 65.01 |
| | DartQuant | 73.24 | 46.26 | 80.41 | 44.80 | 76.09 | 78.17 | 78.24 | 52.65 | 60.37 | 65.58 |
| 4-4-16 | RTN | 50.59 | 34.95 | 52.88 | 24.40 | 5.53 | 31.16 | 29.92 | 22.10 | 23.29 | 30.54 |
| | SmoothQuant | 49.41 | 34.60 | 50.87 | 27.20 | 0.06 | 26.44 | 26.6 | 26.96 | 23.77 | 29.55 |
| | GPTQ | 49.41 | 34.54 | 55.33 | 28.80 | 13.35 | 36.14 | 35.44 | 25.26 | 22.93 | 33.47 |
| | OmniQuant | 49.57 | 33.47 | 53.92 | 26.40 | 4.04 | 36.64 | 30.09 | 24.15 | 22.97 | 31.25 |
| | QuaRot | 66.54 | 43.50 | 76.01 | 38.40 | 66.54 | 72.69 | 67.26 | 43.26 | 49.59 | 58.20 |
| | SpinQuant | 69.46 | 44.73 | 77.20 | 43.40 | 72.77 | 75.90 | 74.16 | 47.61 | 55.38 | 62.29 |
| | OSTQuant | 68.43 | 44.78 | 77.91 | 41.40 | 72.68 | 75.49 | 75.63 | 48.55 | 54.74 | 62.18 |
| | DartQuant | 70.96 | 45.34 | 79.16 | 43.40 | 72.39 | 75.81 | 74.45 | 48.21 | 55.46 | 62.80 |
| 4-4-4 | RTN | 50.59 | 34.95 | 52.88 | 24.40 | 5.53 | 31.16 | 29.92 | 22.10 | 23.29 | 30.54 |
| | GPTQ | 49.17 | 34.70 | 54.62 | 27.00 | 7.18 | 31.48 | 33.92 | 23.21 | 23.14 | 31.60 |
| | QuaRot | 66.77 | 44.78 | 73.61 | 41.00 | 64.93 | 71.69 | 66.33 | 40.02 | 46.77 | 57.32 |
| | SpinQuant | 67.56 | 43.71 | 77.26 | 41.40 | 72.06 | 75.50 | 73.48 | 47.95 | 53.20 | 61.35 |
| | OSTQuant | 68.03 | 44.32 | 77.58 | 42.00 | 71.96 | 74.94 | 74.28 | 46.67 | 54.35 | 61.57 |
| | DartQuant | 70.68 | 44.68 | 78.18 | 43.00 | 72.58 | 75.28 | 75.52 | 48.63 | 54.31 | 62.38 |

Table 13: Comprehensive comparison of Perplexity scores for LLaMA-3 8B across three datasets.

| Bits (W-A-KV) | Method | Wiki | PTB | C4 | Avg |
|---|---|---|---|---|---|
| 16-16-16 | Full Precision | 6.14 | 11.18 | 9.45 | 8.92 |
| 4-8-16 | RTN | 7.20 | 12.65 | 11.19 | 10.35 |
| | SmoothQuant | 84.37 | 139.98 | 113.02 | 112.46 |
| | GPTQ | 7.22 | 12.88 | 10.71 | 10.27 |
| | OmniQuant | 7.17 | 12.68 | 11.6 | 10.48 |
| | QuaRot | 6.55 | 11.74 | 10.48 | 9.59 |
| | SpinQuant | 6.49 | 11.63 | 10.31 | 9.48 |
| | DartQuant | 6.50 | 11.68 | 10.29 | 9.49 |
| 4-4-16 | RTN | 179.40 | 244.36 | 177.91 | 200.56 |
| | SmoothQuant | 2623.41 | 2213.76 | 1811.66 | 2216.28 |
| | GPTQ | 137.12 | 258.71 | 415.63 | 270.49 |
| | OmniQuant | 124.75 | 248.78 | 184.54 | 186.02 |
| | QuaRot | 8.05 | 13.99 | 13.19 | 11.74 |
| | SpinQuant | 7.22 | 12.88 | 11.90 | 10.67 |
| | OSTQuant | 7.26 | 12.81 | 11.9 | 10.66 |
| | DartQuant | 7.32 | 12.51 | 11.81 | 10.58 |
| 4-4-4 | RTN | 354.37 | 372.86 | 333.09 | 353.44 |
| | GPTQ | 282.29 | 308.06 | 900.45 | 496.93 |
| | QuaRot | 8.40 | 14.68 | 13.79 | 12.29 |
| | SpinQuant | 7.41 | 13.33 | 12.23 | 10.99 |
| | OSTQuant | 7.26 | 12.81 | 1.90 | 10.66 |
| | DartQuant | 7.43 | 12.89 | 12.02 | 10.78 |

Table 14: Comprehensive comparison of the average accuracy of LLaMA-3 70B on nine Zero-Shot Commonsense Reasoning tasks.

| Bits (W-A-KV) | Method | WG | SIQA | PIQA | OBQA | LAMB | HS | ARC-E | ARC-C | MMLU | Avg |
|---|---|---|---|---|---|---|---|---|---|---|---|
| 16-16-16 | Full Precision | 80.82 | 50.61 | 84.44 | 48.40 | 79.31 | 84.99 | 85.98 | 64.25 | 75.47 | 72.70 |
| 4-8-16 | RTN | 73.56 | 47.80 | 81.45 | 43.80 | 72.93 | 82.35 | 78.70 | 55.03 | 69.00 | 67.18 |
| | SmoothQuant | 49.25 | 34.95 | 61.92 | 27.20 | 2.02 | 39.39 | 40.49 | 27.39 | 22.95 | 33.95 |
| | GPTQ | 79.48 | 48.36 | 82.37 | 46.60 | 75.30 | 83.18 | 80.30 | 57.08 | 72.28 | 69.44 |
| | OmniQuant | 57.93 | 46.01 | 79.60 | 34.60 | 58.39 | 80.02 | 74.96 | 53.41 | 54.54 | 59.94 |
| | QuaRot | 79.48 | 48.87 | 83.30 | 48.20 | 79.14 | 83.29 | 82.41 | 59.13 | 72.92 | 70.75 |
| | SpinQuant | 80.27 | 49.90 | 84.22 | 49.20 | 78.58 | 84.42 | 83.54 | 61.77 | 73.97 | 71.76 |
| | DartQuant | 80.35 | 49.80 | 83.95 | 48.40 | 78.79 | 84.89 | 85.06 | 61.18 | 73.96 | 71.82 |
| 4-4-16 | RTN | 52.01 | 33.37 | 55.22 | 28.00 | 1.44 | 27.11 | 31.4 | 26.45 | 24.07 | 31.01 |
| | SmoothQuant | 51.62 | 33.42 | 52.12 | 27.40 | 0.00 | 25.68 | 26.01 | 24.15 | 23.28 | 29.30 |
| | GPTQ | 48.07 | 34.70 | 60.72 | 25.40 | 3.53 | 28.77 | 39.81 | 26.96 | 24.77 | 32.53 |
| | OmniQuant | 49.57 | 32.91 | 49.51 | 27.60 | 0.00 | 25.04 | 25.08 | 22.70 | 22.95 | 28.37 |
| | QuaRot | 69.69 | 43.71 | 78.02 | 44.40 | 71.96 | 75.95 | 71.00 | 47.18 | 58.61 | 62.28 |
| | SpinQuant | 73.56 | 44.52 | 79.54 | 46.00 | 74.13 | 81.00 | 79.42 | 56.23 | 60.13 | 66.06 |
| | OSTQuant | 73.64 | 46.32 | 81.56 | 45.40 | 75.61 | 83.19 | 80.13 | 57.17 | 69.29 | 67.94 |
| | DartQuant | 77.27 | 47.54 | 83.08 | 48.00 | 76.44 | 83.61 | 81.57 | 58.02 | 68.96 | 69.39 |
| 4-4-4 | RTN | 48.70 | 33.37 | 51.96 | 30.80 | 1.57 | 26.73 | 31.44 | 25.09 | 24.01 | 30.41 |
| | GPTQ | 50.36 | 34.19 | 58.60 | 30.00 | 3.45 | 28.98 | 39.27 | 27.82 | 24.11 | 32.98 |
| | QuaRot | 70.24 | 43.96 | 79.00 | 39.60 | 69.90 | 75.56 | 71.04 | 48.12 | 56.04 | 61.50 |
| | SpinQuant | 73.80 | 43.81 | 79.38 | 43.40 | 72.56 | 81.20 | 76.81 | 54.10 | 57.81 | 64.76 |
| | OSTQuant | 73.48 | 45.55 | 81.99 | 45.40 | 75.78 | 83.06 | 80.05 | 56.31 | 68.97 | 67.84 |
| | DartQuant | 77.19 | 47.13 | 81.88 | 47.40 | 76.36 | 83.53 | 80.81 | 58.70 | 68.44 | 69.05 |

Table 15: Comprehensive comparison of Perplexity scores for LLaMA-3 70B across three datasets.

| Bits (W-A-KV) | Method | Wiki | PTB | C4 | Avg |
|---|---|---|---|---|---|
| 16-16-16 | Full Precision | 2.86 | 8.53 | 7.17 | 6.19 |
| 4-8-16 | RTN | 5.29 | 10.81 | 21.05 | 12.38 |
| | SmoothQuant | 429.54 | 691.87 | 512.64 | 544.68 |
| | GPTQ | 3.56 | 8.99 | 8.11 | 6.89 |
| | OmniQuant | 8.09 | 21.83 | 14.93 | 14.95 |
| | QuaRot | 3.78 | 9.05 | 7.94 | 6.92 |
| | SpinQuant | 3.46 | 8.76 | 7.66 | 6.63 |
| | DartQuant | 3.45 | 8.84 | 7.69 | 6.66 |
| 4-4-16 | RTN | 28070.21 | 4096.69 | 20005.66 | 17390.85 |
| | SmoothQuant | 5328.88 | 7907.32 | 5491.67 | 6242.62 |
| | GPTQ | 14637.05 | 3119.77 | 24848.08 | 14201.63 |
| | OmniQuant | 363.42 | 462.06 | 317.33 | 380.94 |
| | QuaRot | 6.51 | 12.97 | 12.70 | 10.73 |
| | SpinQuant | 5.92 | 11.66 | 11.25 | 9.61 |
| | OSTQuant | 4.5 | 9.58 | 8.93 | 7.67 |
| | DartQuant | 4.83 | 9.80 | 9.35 | 7.99 |
| 4-4-4 | RTN | 31807.71 | 3742.77 | 17859.72 | 17803.40 |
| | GPTQ | 15320.82 | 2741.99 | 34022.33 | 17361.71 |
| | QuaRot | 6.79 | 14.01 | 13.33 | 11.38 |
| | SpinQuant | 6.15 | 12.1 | 12.26 | 10.17 |
| | OSTQuant | 4.55 | 9.71 | 9.03 | 7.76 |
| | DartQuant | 4.92 | 9.92 | 9.55 | 8.13 |

# D Sample Size Analysis

Table 16 investigates the impact of sample size on DartQuant's performance. All experiments are performed with a token sampling ratio of 10%. The results show that DartQuant's calibration performance remains robust even with a small dataset.

Table 16: Comparison of DartQuant Calibration Results with Different Sample Sizes.

| Model | Sample | WikiText2 | PTB | C4 | Avg |
|-------|--------|-----------|-----|-----|-----|
| 2 7b | 32 | 5.91 | 42.61 | 8.03 | 18.85 |
|  | 64 | 5.88 | 43.33 | 8.00 | 19.07 |
|  | 128 | 5.92 | 42.63 | 7.99 | 18.85 |
|  | 256 | 5.92 | 42.41 | 8.04 | 18.79 |
| 3 8b | 32 | 7.30 | 12.66 | 11.79 | 10.58 |
|  | 64 | 7.30 | 12.77 | 11.79 | 10.62 |
|  | 128 | 7.29 | 12.71 | 11.85 | 10.62 |
|  | 256 | 7.41 | 12.83 | 11.99 | 10.74 |

# E   Comparison with Mixed Precision Quantization Methods

For a more comprehensive comparison, we selected two mixed precision quantization algorithms, QUIK [51] and Atom [52], and compared them with DartQuant under the **4-4-16** bit setting. It is important to note that, for a fair comparison, we preserved QUIK's feature of protecting the first 256 outlier channels, while quantizing the down projection to 4 bits to ensure consistency with our method. Atom was tested using its default settings. The specific experimental results are shown in Tables 17 and 18.

Table 17: Comparison of accuracy with mixed precision quantization methods on zero-shot tasks.

| Model | Method | WG | SIQA | PIQA | OBQA | LAMB | HS | ARC-E | ARC-C | MMLU | AVG |
|-------|--------|-----|------|------|------|------|-----|-------|-------|------|-----|
| 2-7b | QUIK | 62.12 | 41.20 | 73.67 | 36.80 | 61.81 | 68.41 | 62.25 | 37.46 | 27.76 | 52.39 |
|  | Atom | 64.38 | 43.01 | 76.04 | **39.40** | 69.76 | 70.40 | 70.17 | 41.26 | 34.01 | 56.49 |
|  | DartQuant | **67.17** | **44.93** | **76.93** | 39.00 | **71.65** | **73.76** | **70.96** | **42.41** | **35.66** | **58.05** |
| 2-13b | QUIK | 62.67 | 44.37 | 74.86 | 41.40 | 62.27 | 72.88 | 65.87 | 41.38 | 35.32 | 55.67 |
|  | Atom | 69.04 | 45.26 | 77.94 | 43.60 | 73.94 | 75.81 | 72.62 | 45.44 | 45.07 | 60.97 |
|  | DartQuant | **71.11** | **46.16** | **79.27** | **44.20** | **75.18** | **78.04** | **75.38** | **47.61** | **46.80** | **62.64** |
| 2-70b | QUIK | 68.67 | 44.22 | 77.58 | 42.80 | 64.12 | 74.32 | 68.98 | 46.42 | 46.53 | 59.29 |
|  | Atom | 75.29 | 46.27 | 81.12 | 45.73 | 76.44 | 79.98 | 79.18 | 54.88 | 59.30 | 66.47 |
|  | DartQuant | **77.58** | **48.52** | **82.70** | **48.20** | **79.99** | **82.62** | **81.93** | **57.00** | **62.60** | **69.02** |
| 3-8b | QUIK | 59.59 | 39.00 | 65.78 | 35.00 | 42.81 | 58.45 | 52.53 | 34.98 | 29.01 | 46.35 |
|  | Atom | 68.67 | 43.06 | 76.88 | 42.00 | 70.41 | 73.26 | 72.36 | 46.35 | 53.28 | 60.70 |
|  | DartQuant | **70.96** | **45.34** | **79.16** | **43.40** | **72.39** | **75.81** | **74.45** | **48.21** | **55.46** | **62.80** |
| 3-70b | QUIK | 56.83 | 40.94 | 71.33 | 34.80 | 55.07 | 70.11 | 62.16 | 38.82 | 31.87 | 51.33 |
|  | Atom | 74.16 | 44.98 | 78.61 | 45.54 | 72.17 | 76.88 | 74.23 | 51.78 | 61.62 | 64.44 |
|  | DartQuant | **77.27** | **47.54** | **83.08** | **48.00** | **76.44** | **83.61** | **81.57** | **58.02** | **68.96** | **69.39** |

Table 18: Comparison of perplexity with mixed precision quantization methods on the WikiText2, PTB, and C4 datasets.

| Model | Method | WIKI | PTB | C4 | AVG |
|-------|--------|------|-----|-----|-----|
| 2-7b | QUIK | 8.05 | 51.97 | 10.12 | 23.38 |
|  | Atom | 6.03 | 46.77 | 8.25 | 20.35 |
|  | DartQuant | **5.88** | **41.72** | **7.99** | **18.53** |
| 2-13b | QUIK | 7.29 | 65.72 | 9.23 | 27.41 |
|  | Atom | 5.26 | **52.95** | 7.33 | **21.85** |
|  | DartQuant | **5.22** | 54.82 | **7.28** | 22.44 |
| 2-70b | QUIK | 6.36 | 35.28 | 8.77 | 16.80 |
|  | Atom | 3.68 | 28.21 | 6.05 | 12.65 |
|  | DartQuant | **3.64** | **24.90** | **5.99** | **11.51** |
| 3-8b | QUIK | 18.01 | 37.42 | 14.72 | 23.38 |
|  | Atom | 7.57 | 16.67 | 13.28 | 12.50 |
|  | DartQuant | **7.32** | **12.51** | **11.81** | **10.58** |
| 3-70b | QUIK | 10.32 | 21.48 | 17.24 | 16.35 |
|  | Atom | 5.23 | 13.00 | 11.48 | 9.90 |
|  | DartQuant | **4.83** | **9.80** | **9.35** | **7.99** |

Even though DartQuant strictly quantizes all activations and weights to 4 bits (resulting in a lower average bit-width compared to QUIK and Atom), it still achieves accuracy improvements on most datasets. This underscores the effectiveness of our method.

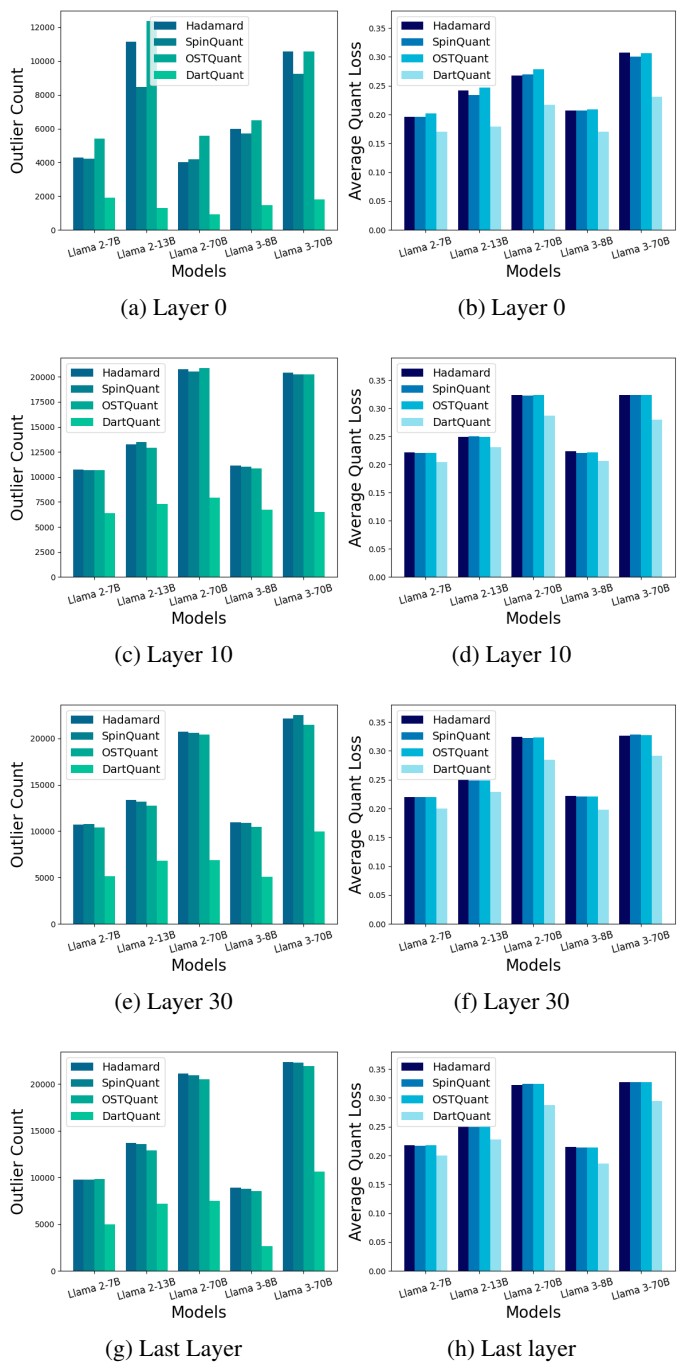

Figure 10: Impact of different transformations on 1000 activations across layers in different models.

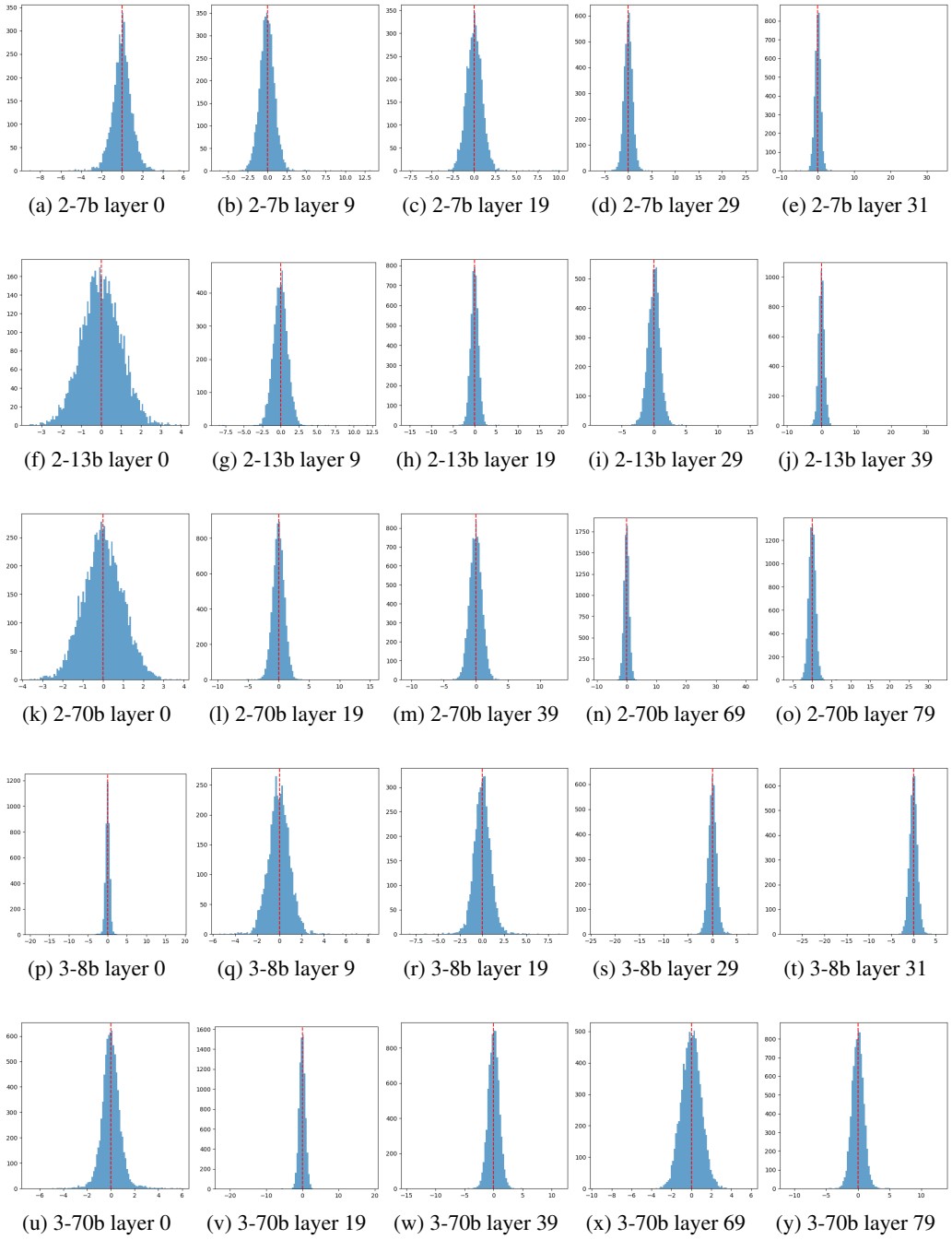

Figure 11: Activation distribution histograms for different layers of various models. The x-axis represents activation values, while the y-axis denotes the channel count.

## F  Effects of Different Transformations on Activation Outliers and Quantization Errors

We extracted 1000 activation samples from different layers of Llama-2 (7B/13B/70B) and Llama-3 (8B/70B) models and analyzed the number of outliers and the average quantization error after applying different transformation methods. The statistical results are shown in Figure 10. As observed, the activations after DartQuant transformation exhibit a significant reduction in outliers, with a marked decrease in quantization loss, further validating the effectiveness of DartQuant in large-scale language model quantization.

## G  Activation Distribution

We conducted an in-depth investigation into the distribution characteristics of activations in LLMs. First, we randomly sampled 1,000 activation samples from each model and computed their mean, variance and kurtosis. The detailed statistical results are presented in Table 19. As observed, the mean of activations is close to zero, the variance is approximately 1, and the activations exhibit high kurtosis (whereas the kurtosis of a Gaussian distribution is 0). These statistics indicate that most activation values are concentrated around zero and exhibit significant heavy-tailed properties.

Figure 11 presents the activation distribution histograms across different layers of various models. As shown, apart from a few outliers, the overall activation distribution is symmetric around zero. These distribution characteristics closely align with those of a Laplacian distribution. Therefore, we model the activation distribution as a simplified Laplacian distribution.

Table 19: Statistics of each model activation.

| Model | Kurtosis | Mean | Variance |
|---|---|---|---|
| Llama 2-7b | 87.69 | 1.18e-02 | 9.97e-01 |
| Llama 2-13b | 58.99 | 3.17e-03 | 9.98e-01 |
| Llama 2-70b | 245.10 | -4.88e-03 | 9.97e-01 |
| Llama 3-8b | 44.32 | -2.92e-05 | 9.91e-01 |
| Llama 3-70b | 37.35 | 4.64e-03 | 9.80e-01 |

## H  Performance on MoE

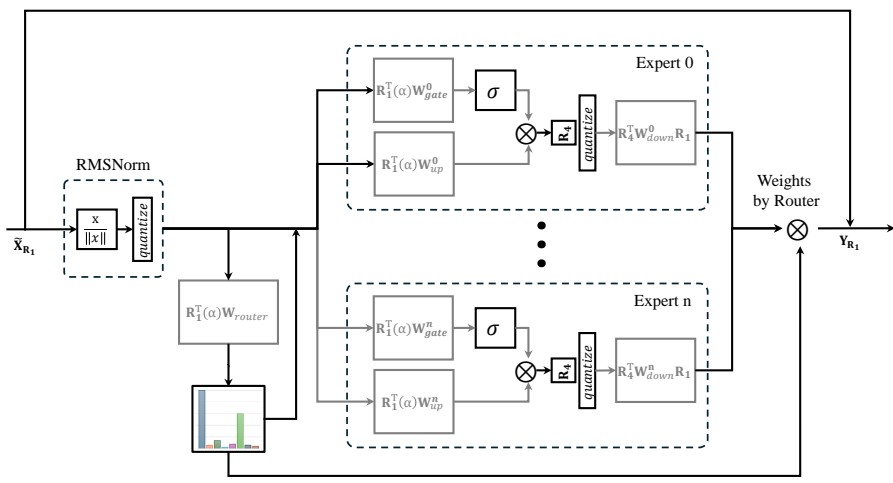

Figure 12: MoE applied in DartQuant. The fusion method of the MHSA block is the same as that of dense models 9.

Table 20: Zero-shot and perplexity evaluation results of Mixtral-7×8B using DartQuant.

| Method | Bits | WG | SIQA | PIQA | OBQA | LAMB | HS | ARC-E | ARC-C | MMLU | AVG | WIKI |
|---|---|---|---|---|---|---|---|---|---|---|---|---|
| Baseline | FP16 | 76.24 | 49.80 | 83.57 | 47.60 | 78.05 | 84.03 | 83.67 | 59.30 | 67.29 | 69.95 | 3.84 |
| RTN | | 47.28 | 34.49 | 53.16 | 25.60 | 0.41 | 29.13 | 34.39 | 24.66 | 23.57 | 30.30 | 345.40 |
| GPTQ | 4-4-16 | 50.51 | 34.70 | 52.88 | 27.40 | 1.69 | 29.81 | 33.63 | 25.34 | 23.30 | 31.03 | 219.61 |
| Quarot | | 69.46 | 45.45 | 79.60 | 44.00 | 74.27 | 79.01 | 76.47 | 53.41 | 59.42 | 64.57 | 4.97 |
| DartQuant | | **71.90** | **46.09** | **80.47** | **45.20** | **75.02** | **80.19** | **77.31** | **54.10** | **59.93** | **65.58** | **4.75** |
| RTN | | 52.09 | 34.14 | 53.26 | 24.00 | 0.29 | 29.43 | 34.64 | 23.72 | 23.25 | 30.54 | 329.05 |
| GPTQ | 4-4-4 | 49.17 | 33.52 | 54.03 | 26.40 | 1.69 | 29.76 | 32.79 | 24.66 | 23.29 | 30.59 | 239.59 |
| Quarot | | 67.32 | 45.45 | 80.14 | 42.40 | 71.98 | 78.74 | 75.76 | 52.05 | 58.44 | 63.59 | 5.03 |
| DartQuant | | **69.85** | **46.11** | **80.29** | **45.12** | **73.47** | **79.98** | **77.61** | **52.13** | **60.35** | **64.99** | **4.80** |

Table 21: Zero-shot and perplexity evaluation results of DeepSeek-moe-16b-base using DartQuant.

| Method | Bits | WG | SIQA | PIQA | OBQA | LAMB | HS | ARC-E | ARC-C | MMLU | AVG | WIKI |
|---|---|---|---|---|---|---|---|---|---|---|---|---|
| Baseline | FP16 | 69.93 | 45.96 | 80.52 | 43.20 | 72.99 | 77.43 | 73.19 | 47.53 | 38.18 | 60.99 | 6.51 |
| RTN | | 52.17 | 35.93 | 67.79 | 31.60 | 32.51 | 48.38 | 53.58 | 33.70 | 24.13 | 42.20 | 251.42 |
| GPTQ | 4-4-16 | 52.41 | 38.54 | 68.88 | 33.80 | 36.66 | 50.22 | 57.66 | 33.36 | 23.95 | 43.94 | 103.99 |
| Quarot | | 66.64 | 45.55 | 78.40 | 42.60 | 70.81 | 74.68 | 70.34 | 44.11 | 34.05 | 58.58 | 7.02 |
| DartQuant | | **67.72** | **45.70** | **78.89** | **43.00** | **72.44** | **75.97** | **71.13** | **45.23** | **35.04** | **59.46** | **6.88** |
| RTN | | 49.72 | 36.34 | 65.18 | 31.60 | 27.40 | 44.77 | 51.05 | 30.46 | 23.64 | 40.02 | 276.81 |
| GPTQ | 4-4-4 | 52.25 | 35.62 | 65.40 | 30.20 | 29.23 | 46.89 | 54.25 | 30.72 | 24.41 | 41.00 | 127.74 |
| Quarot | | 66.32 | 44.83 | 78.16 | 42.32 | 70.53 | 74.68 | 70.10 | 43.94 | 33.54 | 58.27 | 7.07 |
| DartQuant | | **67.24** | **45.20** | **78.67** | **42.80** | **71.46** | **75.68** | **71.02** | **44.34** | **35.06** | **59.05** | **6.91** |

To demonstrate the adaptability of DartQuant to different model architectures, we also applied it to two MoE models (Mixtral-7×8B and DeepSeek-MoE-16B). The rotation matrix is integrated with the MoE architecture as shown in Figure 12. As shown in Table 20 and 21, DartQuant continues to demonstrate outstanding performance on the MoE architecture, with improved accuracy compared to QuaRot, which uses random rotations.

# I Additional Ablation Analysis of Whip Loss

To further demonstrate the effectiveness of the Whip loss, we evaluated PPL and zero-shot accuracy under different loss functions. As shown in Table 22, the results highlight the advantages of Whip loss in both PPL and zero-shot accuracy.

Table 22: Comparison of zero-shot accuracy and perplexity across different loss functions.

| model | loss | WG | ARC-E | ARC-C | MMLU | WIKI | PTB | C4 |
|---|---|---|---|---|---|---|---|---|
| Llama 2-7b | Quant | 66.30 | 68.90 | 41.72 | 33.73 | 6.03 | 45.70 | 8.17 |
| | Variance | 66.85 | 68.52 | 41.30 | 34.70 | 6.03 | 44.96 | 8.20 |
| | Kurtosis | 66.22 | 69.87 | 42.15 | 35.64 | 5.99 | 46.57 | 8.14 |
| | Whip | **67.17** | **70.96** | **42.41** | **35.66** | **5.90** | **42.94** | **8.01** |
| Llama 3-8b | Quant | 68.19 | 71.46 | 46.33 | 54.08 | 7.64 | 13.17 | 12.32 |
| | Variance | 69.61 | 72.52 | 46.93 | 53.30 | 7.66 | 12.92 | 12.32 |
| | Kurtosis | 69.93 | 71.72 | 45.99 | 54.16 | 7.64 | 13.09 | 12.37 |
| | Whip | **70.96** | **74.45** | **48.21** | **55.46** | **7.29** | **12.71** | **11.85** |

# J The Mechanism of Whip Loss

As shown in Figure 5, the smoothing of matrix peaks is not solely due to the amplification of small values near zero; it also involves a broader redistribution of the activation values.

Specifically, the Whip Loss we designed pushes small activation values away from zero, making them more evenly distributed, which in turn helps reduce quantization error. Under the norm-invariance constraint imposed by the rotation transformation, activation values are effectively redistributed: high-peak regions in the distribution histogram are compressed, while low-value regions are filled, leading to an overall "peak smoothing" effect.

Thus, the smoothing of matrix peaks is not merely a result of amplifying small values but also stems from the uniformization of the overall distribution—one of the key objectives of DartQuant.

The property of norm invariance originates from the mathematical characteristics of rotation matrices. Let $W$ be a rotation matrix (i.e., an orthogonal matrix) satisfying $W^T W = I$. For any input vector $x$, the transformed norm follows:

$$\|Wx\|_2 = (Wx)^T(Wx) = x^T W^T W x = x^T x = \|x\|_2. \tag{6}$$

This confirms that rotation preserves the Euclidean norm of the vector.

o illustrate this effect, we provide a four-dimensional example explaining why amplifying small values under the norm-invariance constraint leads to outlier reduction. Consider a vector $x = [x_1, x_2, x_3, x_4]$, where $x_1, x_2$ and $x_3$ have absolute values close to zero, while $x_4$ is an outlier. Introducing a perturbation $\epsilon_i (i = 1, 2, 3, 4.)$ to each component, the new norm of the vector becomes:

$$\|\tilde{x}\|_2 = \sqrt{\tilde{x}_1^2 + \tilde{x}_2^2 + \tilde{x}_3^2 + \tilde{x}_4^2} \tag{7}$$

$$= \sqrt{(x_1 + \varepsilon_1)^2 + (x_2 + \varepsilon_2)^2 + (x_3 + \varepsilon_3)^2 + (x_4 + \varepsilon_4)^2} \tag{8}$$

$$= \sqrt{x_1^2 + x_2^2 + x_3^2 + x_4^2} = \|x\|_2. \tag{9}$$

Clearly, if $\epsilon_1, \epsilon_2, \epsilon_3 > 0$, it must follow that $\epsilon_4 < 0$. DartQuant effectively leverages this constraint to suppress outliers.

## K   Hyperparameter Settings

The specific hyperparameter settings for DartQuant are shown in Table 23. It is important to note that the latent parameter $Z_0$ is initialized using a random Hadamard matrix.

Table 23: Comparison of zero-shot accuracy and perplexity across different loss functions.

| Rotation | Model | LR | Epoch | Optimizer | BS |
|---|---|---|---|---|---|
| | 2-7b | 2.00E-03 | 10 | SGD | 64 |
| | 2-13b | 1.00E-02 | 10 | SGD | 64 |
| R1 | 2-70b | 1.00E-03 | 10 | SGD | 64 |
| | 3-8b | 8.00E-03 | 10 | SGD | 64 |
| | 3-70b | 3.00E-03 | 10 | SGD | 64 |
| R2 | All | 1.00E-03 | 10 | SGD | 64 |

## L   Limitation

DartQuant is tailored for uniformly distributed integer formats. Its effectiveness on alternative formats, such as FP4, or other non-uniform numerical representations, remains to be further explored and validated. Furthermore, Whip assumes that activations are approximately zero-mean, which generally holds true for most transformer layers. However, in rare cases where the activation mean significantly deviates from zero, the effectiveness of Whip may degrade.

In the future, we can explore a wider range of distribution transformation methods to better accommodate various numerical formats.

## M   Impact Statement

This work introduces DartQuant, an efficient rotational distribution calibration method for LLM quantization. DartQuant simplifies the optimization of rotation matrices while avoiding the overfitting risks associated with end-to-end training. It successfully quantizes a 70B model on a single RTX 3090, marking a significant advancement. This progress not only improves the efficiency of LLMs in practical applications but also provides a practical solution for large-scale model quantization, contributing to the accessibility and scalability of AI technologies. However, if misused for harmful purposes, it could have a negative social impact.

