# OpenReview forum: "DartQuant: Efficient Rotational Distribution Calibration for LLM Quantization"
_NeurIPS.cc/2025/Conference — NeurIPS 2025 poster_

### Official Review · Reviewer_uEAv · 2025-06-03

**Clarity:** 3
**Significance:** 2
**Originality:** 3
**Rating:** 3
**Confidence:** 4

**Summary:**

DartQuant is a method for efficient rotational calibration in LLM quantization. By replacing costly end-to-end fine-tuning of rotation matrices with a distribution-aware calibration approach, DartQuant reduces the risk of overfitting and dramatically lowers computational and memory costs during compression.

**Questions:**

1. What is the metrics of outlier in Figure 3? It should be included in the article.
2. Which baseline is used in Table 1?  It should be stated in the article.
3. Line 283 states "In the main results, we apply GPTQ to rotate the weights.", but GPTQ doesn't have rotate. I guess it's "reconstruct"?
4. Why could DartQuant avoid overfitting? It still use calibration data for both rotate and reconstruct (if above is true).
5. Line 25 is wrong. lora doesn't improve inference efficiency.

**Ethical Concerns:**

["NO or VERY MINOR ethics concerns only"]

**Final Justification:**

From my perspective, the contribution is not compelling. The primary improvement is compression time, which is relatively minor given existing fast PTQ methods. For PTQ, the key metrics are (i) performance, (ii) inference speed, (iii)inference memory, and (iv) compression speed, with the last being least important. In addition, the manuscript contains several minor errors. Overall, I do not find the work strong enough for acceptance.

**Limitations:**

Limitation in appendix.

**Paper Formatting Concerns:**

NaN

**Quality:**

3

**Strengths And Weaknesses:**

Strengths:
1. The overall design of Whip loss makes sense. Because the norm is constant, reducing small-value would also reduce outliers.
2. The use of autograd qr decomposition makes sense, which avoids the time-consuming Cayley SGD but still guarantee the orthogonality. But I'm not sure if the gradient of qr decomposition is stable or not.

Weakness:
1. The main weakness is that the improvement is not significant. The performance is very marginal compared to SpinQuant. The improvement on compression time and memory is large. But I doubt whether compression time is an important objective in quantization, compared with time and memory during inference, or performance.
2. In four rotation matrix R1 ... R4, only R1 and R2 are learned by DartQuant, while R3 and R4 use Hadamard rotations like previous work, which limits the value of this method. By the way, the Hadamard matrix of R3 and R4 is random, right?
3. See "Questions".

---

> ### Author Rebuttal · Authors · 2025-07-30
>
> We are grateful for your review and positive feedback. We will address each of the weaknesses and questions raised (in a combined way) in detail below. We will be happy to answer any further questions.
>
> **Strengths2:** *But I'm not sure if the gradient of qr decomposition is stable or not.*
>
>  Thank you for the comment. Regarding the gradient stability of QR decomposition, we provided empirical analysis in our reply to SzJU Question4. Under various initializations and hyperparameters, the autograd-based QR-Ortho consistently converged and showed strong robustness to learning rate. Hence, it is stable in practice.
>
> |ID|LR|FinalObjective|Iterations|
> |-|-|-|-|
> |1|1.00E-03|2004.9094|138|
> |2|1.00E-03|2004.9535|141|
> |3|1.00E-03|2004.9191|140|
> |4|2.00E-03|2004.9507|131|
> |5|5.00E-03|2004.8560|123|
>
> **Weakness1:** *The main weakness is that the improvement is not significant. The performance is very marginal compared to SpinQuant. The improvement on compression time and memory is large. But I doubt whether compression time is an important objective in quantization, compared with time and memory during inference, or performance.*
>
> Thanks for the feedback. First, we would like to highlight the practical importance of compression time in post-training quantization (PTQ), especially as model sizes and deployment scales continue to grow. Our primary contribution is improving the efficiency of rotation matrix optimization, a point that has also been positively noted by reviewers sxB6 and SzJU.
>
> For example, SpinQuant requires over 40 GPU hours (8×A800) to optimize LLaMA 2-70B. As models continue to scale (e.g., LLaMA 3.1 405B), and development cycles accelerate, optimization cost becomes an increasingly important factor in the applicability of PTQ techniques.
>
> This trend is also observable in practice: although some existing methods can achieve strong accuracy, their relatively high computational cost can limit practical adoption. In contrast, methods with lower overhead (e.g., AutoGPTQ) have seen broader use in the community, due to their balance between speed and effectiveness. Similarly, while SpinQuant provides strong accuracy, its higher resource cost may affect its ease of use in large-scale or time-sensitive settings.
>
> Another point we would like to emphasize is that,  DartQuant supports further accuracy improvements by increasing calibration data size, especially when extreme efficiency is not the primary concern. For instance, using 512 calibration samples on 7B/8B models, the optimization time remains moderate (about 24.97 minutes), approaching the runtime of SpinQuant while offering higher accuracy. The experimental results are as follows:
>
> |Method|Model|WIKI|ARC_C|ARC_E|OBQA|PIQA|WG|AVG|
> |-|-|-|-|-|-|-|-|-|
> |Baseline|2 7B|5.47|46.33|74.54|44.20|79.05|69.06|62.64|
> |SpinQuant||5.99|42.32|70.50|40.60|76.44|64.17|58.81|
> |Ours(paper)||5.93|44.20|69.87|38.60|77.48|66.69|59.37|
> |Our(512)||5.72|45.09|72.72|43.40|78.18|68.98|61.67|
> |Baseline|3 8B|6.14|53.24|77.82|45.00|80.96|73.32|66.07|
> |SpinQuant||7.41|47.95|73.48|41.40|77.26|67.56|61.53|
> |Ours(paper)||7.43|48.63|75.52|43.00|78.18|70.68|63.20|
> |Our(512)||6.84|51.02|76.47|44.00|79.27|71.67|64.49|
>
> From the results, it can be observed that our method achieves a significant accuracy improvement over SpinQuant when the optimization time is aligned. In summary, we believe that DartQuant is more practical for deployment with a small calibration set due to its high optimization efficiency. On the other hand, in scenarios where optimization time is not a major concern, our method can achieve outstanding accuracy. If you have any further questions regarding this, please do not hesitate to let us know.
>
> **Weakness2:** *In four rotation matrix R1 ... R4, only R1 and R2 are learned by DartQuant, while R3 and R4 use Hadamard rotations like previous work, which limits the value of this method. By the way, the Hadamard matrix of R3 and R4 is random, right?*
>
> In current experiments, we apply DartQuant to R1 and R2, and use random Hadamard for R3 and R4 to minimize inference latency, following SpinQuant and OSTQuant.
>
> However, DartQuant is also compatible with R3/R4 optimization. Below is the accuracy improvement when optimizing R4 with DartQuant.
>
> |Method|Model|WIKI|ARC_C|ARC_E|OBQA|PIQA|WG|AVG|
> |-|-|-|-|-|-|-|-|-|
> |baseline|2 7B|5.47|46.33|74.54|44.20|79.05|69.06|62.64|
> |dartquant||5.93|44.20|69.87|38.60|77.48|66.69|59.37|
> |dartquant+r4||5.92|44.11|71.04|42.60|77.64|67.80|60.64|
> |baseline|3 8B|6.14|53.24|77.82|45.00|80.96|73.32|66.07|
> |dartquant||7.43|48.63|75.52|43.00|78.18|70.68|63.20|
> |dartquant+r4||7.35|49.57|75.84|43.40|78.94|72.85|64.12|
>
> **Question1:** *What is the metrics of outlier in Figure 3? It should be included in the article.*
>
> Outliers in Fig.3 follow PrefixQuant[1]: values >3×median (before rotation). We will clarify this definition in the appendix.
>
> [1] Chen, Mengzhao, et al. "Prefixquant: Static quantization beats dynamic through prefixed outliers in llms." (2024).
>
> **Question2:** *Which baseline is used in Table 1? It should be stated in the article.*
>
> The baseline in Table 1 is consistent with Table 2 and detailed in Appendix C. We will clarify this in Table 1 to avoid confusion.
>
> **Question3:** *Line 283 states "In the main results, we apply GPTQ to rotate the weights.", but GPTQ doesn't have rotate. I guess it's "reconstruct"?*
>
> You are absolutely correct that Line 283 contains a misstatement. GPTQ does not involve weight rotation; rather, it reconstructs weights via quantization-aware optimization. We will revise the wording to accurately reflect that GPTQ applies reconstruction, not rotation.
>
> **Question4:** *Why could DartQuant avoid overfitting? It still use calibration data for both rotate and reconstruct (if above is true).*
>
> DartQuant uses calibration data, but unlike end-to-end methods, it optimizes a proxy target: distribution alignment via Whip loss, not final quantization error. This indirect optimization reduces overfitting to calibration data. As shown in Table 5, performance remains stable across calibration sizes, with no “diagonal effect.” Thus, the distribution-alignment design helps DartQuant avoid overfitting seen in SpinQuant.
>
> **Qusetion5:** *Line 25 is wrong. lora doesn't improve inference efficiency.*
>
> We acknowledge the inaccuracy in Line 25: while LoRA is effective in reducing fine-tuning cost and memory footprint, it does not bring improvements to inference efficiency. We will amend this reference to prevent any misunderstanding.
>
> We appreciate your attention to detail and your valuable suggestions, which help us improve the clarity and accuracy of our work.

---

> > ### Comment · Reviewer_uEAv · 2025-08-04
> >
> > Thanks for the reply. I would remain the score.

---

> > > ### Author Response · Authors · 2025-08-04
> > >
> > > Dear Reviewer uEAv,
> > >
> > > Thank you very much for your follow-up and for taking the time to review our paper. We respect your decision to maintain your current score.
> > >
> > > If you have any further questions or suggestions, we would be more than happy to address them. Your feedback is greatly appreciated and will help us continue to improve this work.

---

> > > > ### Comment · Reviewer_uEAv · 2025-08-06
> > > >
> > > > I appreciate the authors' response and supplementary experiments. However, I remain unconvinced that the main contribution is an improvement in compression efficiency. Although DartQuant shows shorter compression time than SpinQuant and OSTQuant, its reliance on multi-iteration gradient descent is still computationally heavy and, in my view, misaligned with the stated efficiency goal. Accordingly, my assessment is unchanged and I will keep my score.

---

> > > > > ### Author Response · Authors · 2025-08-06
> > > > >
> > > > > We sincerely thank the reviewer for their continued engagement. We would like to respectfully clarify two key aspects that we believe directly address the reviewer’s concern regarding DartQuant’s contribution to compression efficiency: (1) concrete improvements in resource efficiency, and (2) clear gains in quantization accuracy under comparable settings.
> > > > >
> > > > > **(1) Substantial Compression Efficiency Gains**
> > > > >
> > > > > We would like to emphasize that the term "efficient" here refers to faster speed and less GPU resource consumption during the quantization solving process. Thus, respectfully, the statement of *"Although DartQuant shows shorter compression time than SpinQuant and OSTQuant, its reliance on multi-iteration gradient descent is still computationally heavy and, in my view, misaligned with the stated efficiency goal"* is weird and doesn't make sense to us. One of the main contribution of DartQuant over previous SOTA quantization method such as SpinQuant and OSTQuant is its efficiency, i.e., **shorter compression time**, and **less GPU resource consumption**.
> > > > >
> > > > > **Shorter compression time:** While DartQuant employs a lightweight multi-iteration gradient update, its overall compression cost is significantly lower than prior methods. The multi-iteration gradient descent optimization is conducted layer by layer with fast converge (Figure 7), due to our Whip loss and layer-wise optimization, as well as the QR-orth optimizer. As a result, the **end-to-end compression time** of DartQuant is much shorter than SpinQuant and OSTQuant. The detailed results are reported in Table 3 of our main paper, and we also put it here bellow.
> > > > >
> > > > > | Cost            | Method          | 7B    | 13B    | 70B    |
> > > > > | --------------- | --------------- | ----- | ------ | ------ |
> > > > > | Time (GPU hour) | SpinQuant       | 0.30  | 0.70   | 42.90  |
> > > > > |                 | OSTQuant        | 0.30  | 0.80   | 44.00  |
> > > > > |                 | DartQuant       | 0.14  | 0.23   | 0.91   |
> > > > > |                 | DartQuant(3090) | 0.43  | 0.70   | 2.90   |
> > > > > | Memory (GiB)    | SpinQuant       | 19.98 | 33.73  | 238.89 |
> > > > > |                 | OSTQuant        | 42.25 | 239.16 | 583.86 |
> > > > > |                 | DartQuant       | 17.41 | 21.40  | 23.47  |
> > > > > |                 | DartQuant(3090) | 17.41 | 21.40  | 23.47  |
> > > > >
> > > > >
> > > > > From the results, we can see that, for LLaMA 2 7B, DartQuant reduces GPU-hour consumption by **2.14×** compared to SpinQuant and OSTQuant. For LLaMA 3 70B, these benefits are even more pronounced: DartQuant (0.91 gpu hours) achieves up to **47×/48×** speedup over SpinQuant (42.9 gpu hours) and OSTQuant (44.0 gpu hours).
> > > > >
> > > > > **Less GPU resource consumption:** In addition to the end-2-end speedup, DartQuant substantially reduce the GPU memory consumption compared with SpinQuant or OSTQuant. Take the LLaMA 3 70B for example, DartQuant (23.47 GB) shows **10.2×/24.9×** lower memory usage with respect to SpinQuant (238.89 GB) and OSTQuant (583.86 GB), respectively.
> > > > >
> > > > > Importantly, DartQuant can perform 70B model compression using **a single RTX 3090 GPU**, whereas SpinQuant requires 8×A800 GPUs, which is significantly more expensive and less accessible. This further underscores DartQuant’s efficiency and scalability in realistic hardware settings.
> > > > >
> > > > > These improvements are achieved without any hyperparameter tuning or model-specific adaptations, making DartQuant highly practical for large-scale deployment scenarios, where both time and memory are critical bottlenecks. Contrary to the reviewer’s concern, these results demonstrate that DartQuant substantially reduces actual computation burden, fulfilling our efficiency objective in a practical and scalable manner.

---

> > > > > > ### Author Response · Authors · 2025-08-06
> > > > > >
> > > > > > **(2) Consistent Accuracy Improvements with and without Efficiency Trade-offs**
> > > > > >
> > > > > > Beyond speed and memory, DartQuant also improves quantization accuracy, particularly in low-data settings. The reviewer mentioned concerns regarding compression efficiency, but it is important to note that DartQuant also improves final model quality with the same or even fewer calibration samples.
> > > > > >
> > > > > > Below, we provide extended 4-8-16 and 4-4-4 quantization results using 512 and 800 calibration samples. As shown, DartQuant consistently outperforms both Quarot, SpinQuant and OstQuant in terms of average accuracy:
> > > > > >
> > > > > > **4-8-16**
> > > > > > | Method         | Model | WIKI | ARC_C | ARC_E | OBQA  | PIQA  | WG    | AVG   |
> > > > > > | -------------- | ----- | ---- | ----- | ----- | ----- | ----- | ----- | ----- |
> > > > > > | baseline       | 3 8B  | 6.14 | 53.24 | 77.82 | 45.00 | 80.96 | 73.32 | 66.07 |
> > > > > > | quarot         |       | 6.55 | 51.79 | 77.40 | 43.80 | 79.49 | 74.11 | 65.31 |
> > > > > > | spinquant(800) |       | 6.55 | 51.88 | 77.10 | 44.80 | 79.92 | 74.19 | 65.58 |
> > > > > > | Ours(paper)    |       | 6.50 | 52.65 | 78.24 | 44.80 | 80.41 | 73.24 | 65.87 |
> > > > > > | Ours(512)      |       | 6.31 | 52.98 | 78.13 | 44.80 | 80.65 | 73.28 | 65.96 |
> > > > > > | Ours(800)      |       | 6.30 | 52.88 | 77.61 | 45.00 | 80.09 | 74.43 | 66.00 |
> > > > > >
> > > > > >
> > > > > > AVG accuracy improvement: **+0.42** over SpinQuant, even with equal calibration (800 samples).
> > > > > >
> > > > > > WIKI PPL reduction: from **6.55→6.30**, showing improved language modeling fidelity.
> > > > > >
> > > > > > **4-4-4**
> > > > > > |Method|Model|WIKI|ARC_C|ARC_E|OBQA|PIQA|WG|AVG|
> > > > > > |-|-|-|-|-|-|-|-|-|
> > > > > > |Baseline|38B|6.14|53.24|77.82|45.00|80.96|73.32|66.07|
> > > > > > |quarot|||8.40|40.02|66.33|41.00|73.61|66.77|57.55|
> > > > > > |spinquant(800)|||7.41|47.95|73.48|41.40|77.26|67.56|61.53|
> > > > > > |ostquant(1000)|||7.26|46.67|74.28|42.00|77.58|68.03|61.71|
> > > > > > |Ours(paper)|||7.43|48.63|75.52|43.00|78.18|70.68|63.20|
> > > > > > |Our(512)|||6.84|51.02|76.47|44.00|79.27|71.67|64.49|
> > > > > > |Ours(800)|||6.81|50.91|76.61|44.80|79.22|71.90|64.69|
> > > > > >
> > > > > > AVG accuracy improvement: **+3.16** over SpinQuant with same 800 samples. **+2.98** over OstQuant with 1000 samples.
> > > > > >
> > > > > > WIKI PPL improvement: **7.41 → 6.81**, a **~47%** reduction vs. SpinQuant, **7.26 → 6.81**, a **~40%** reduction vs. OstQuant.
> > > > > >
> > > > > > **Key Takeaways:**
> > > > > >
> > > > > > - With fewer calibration samples (512/128), DartQuant already surpasses SpinQuant (800) in both AVG accuracy and WIKI PPL.
> > > > > > - With equal calibration (800), DartQuant further improves accuracy to 66.00 (4-8-16) and 64.69 (4-4-4), significantly outperforming all baselines.
> > > > > > - Improvements are consistent across tasks and metrics, demonstrating strong robustness and generalization.
> > > > > >
> > > > > > In summary, DartQuant delivers practical and measurable improvements over existing methods in both compression cost (speed + memory) and quantization quality (PPL and zero-shot accuracy). We believe these results demonstrate a clear advance toward efficient LLM compression. We hope this additional evidence alleviates the reviewer’s concerns and would greatly appreciate reconsideration of the overall assessment.

---

> > > > > > > ### Author Response · Authors · 2025-08-08
> > > > > > >
> > > > > > > Dear Reviewer,
> > > > > > >
> > > > > > > There are fewer than 20 hours remaining in the rebuttal discussion phase. We sincerely wanted to check whether our previous responses have addressed your concerns. Please let us know if there are any remaining questions, we would be happy to clarify them promptly.
> > > > > > >
> > > > > > > Thank you again for your time and consideration.

---

### Official Review · Reviewer_gzqM · 2025-06-25

**Clarity:** 3
**Significance:** 2
**Originality:** 4
**Rating:** 4
**Confidence:** 5

**Summary:**

The paper proposes a new approach for finding orthogonal transformations, which is needed for reducing the activation outliers during post-training quantization.

**Questions:**

I have some questions and comments on this paper:

1. I found the idea interesting. The issue with the current scheme is that the improvement is a bit marginal even when we compare against QuaRot which doesn't do anything (no training). However, this is nothing with the paper, but more with the PTQ schemes. It seems that we hit the wall, considering the limits of the 4-bit compression. That's why we cannot expect large improvements in all these kinds of work.

2. Considering the above fact, I think the improvement would be even closer (compare to something like QuaRot) in more recent 4-bit formats. For this, can you please provide an experiment where you compare DartQuant with QuaRot (I do not suggest SpinQuant as I know that will take 4-5 hours to run) on NvFP4? I would improve my score if we get non-trivial improvement over that format.

**Ethical Concerns:**

["NO or VERY MINOR ethics concerns only"]

**Final Justification:**

After a relatively long discussion with the authors, they provided additional data clarifying aspects of the work that were previously unclear to me. As a result, I am convinced to raise my score for this paper.

**Limitations:**

Yes.

**Paper Formatting Concerns:**

The formatting doesn't have any issue

**Quality:**

3

**Strengths And Weaknesses:**

Strengths

- The paper is well-written and easy to follow
- The idea is interesting
- The experiment setup is proper

Weaknesses

- The improvement is somehow marginal (please see the Question section)

---

> ### Author Rebuttal · Authors · 2025-07-30
>
> We sincerely thank the reviewer for the encouraging comments, and we are glad to hear that you found our idea interesting. Below we provide detailed responses to each of your comments and concerns.
>
> **Question1:** *I found the idea interesting. The issue with the current scheme is that the improvement is a bit marginal even when we compare against QuaRot which doesn't do anything (no training). However, this is nothing with the paper, but more with the PTQ schemes. It seems that we hit the wall, considering the limits of the 4-bit compression. That's why we cannot expect large improvements in all these kinds of work.*
>
> We appreciate your valuable observations. While QuaRot indeed serves as a strong baseline for 4-bit PTQ quantization, we believe the improvements introduced by rotation matrix optimization are still meaningful—particularly for more recent and challenging models such as LLaMA 3 8B/70B, 3.2B, or 1B, and in even lower-bit quantization settings.
>
> For instance, in the 4-4-4 setting (Table 2), DartQuant improves the performance of LLaMA 3 8B from QuaRot’s 12.29/57.32 (PPL/Zero-Shot) to 10.78/62.38, and LLaMA 3 70B from 11.38/61.50 to 8.13/69.05. These are non-trivial improvements.
>
> We would also like to emphasize that, in the original paper, we prioritized extreme optimization efficiency by using a very small number of calibration samples. However, if one trades some speed for better accuracy, DartQuant benefits significantly from larger calibration sets (512 samples same as [1], SpinQuant use 800 samples). Detailed results are as follows:
>
> |Method|Model|WIKI|ARC_C|ARC_E|OBQA|PIQA|WG|AVG|
> |-|-|-|-|-|-|-|-|-|
> |baseline|2 7B|5.47|46.33|74.54|44.20|79.05|69.06|62.64|
> |quarot| |6.17|41.64|68.43|40.60|77.04|65.27|58.60|
> |Ours(paper)| |5.93|44.20|69.87|38.60|77.48|66.69|59.37|
> |Ours(512)| |5.72|45.09|72.72|43.40|78.18|68.98|61.67|
> |baseline|3 8B|6.14|53.24|77.82|45.00|80.96|73.32|66.07|
> |quarot| |8.40|40.02|66.33|41.00|73.61|66.77|57.55|
> |Ours(paper)| |7.43|48.63|75.52|43.00|78.18|70.68|63.20|
> |Ours(512)| |6.84|51.02|76.47|44.00|79.27|71.67|64.49|
>
> As shown, under larger calibration sets, DartQuant consistently outperforms QuaRot on both models. In this light, we respectfully suggest that: "Yes, it is possible to expect substantial improvements from our method."
>
> [1] KurTail: Kurtosis-based LLM Quantization, SLLM workshop ICLR 2025
>
> **Question2:** *Considering the above fact, I think the improvement would be even closer (compare to something like QuaRot) in more recent 4-bit formats. For this, can you please provide an experiment where you compare DartQuant with QuaRot (I do not suggest SpinQuant as I know that will take 4-5 hours to run) on NvFP4? I would improve my score if we get non-trivial improvement over that format.*
>
> To further illustrate DartQuant’s advantages in more difficult scenarios, we conducted additional experiments on LLaMA 3.2B and 1B under 4-bit and 3-bit quantization. In line with your suggestion, we also included evaluations under the NvFP4 format.
>
> Specifically, we implemented NvFP4 quantization using the FP4 codebase from the microsoft/microxcaling repository. Due to the use of complex secondary quantization in NvFP4, the inference speed is substantially reduced. Given limited time, we report results only on the two smaller models mentioned above. The quantitative results are presented below.
>
> |Method|Model|WIKI|ARC_C|ARC_E|OBQA|PIQA|WG|AVG|
> |-|-|-|-|-|-|-|-|-|
> |3.2 3B|baseline|7.81|46.07|71.63|43.40|77.52|69.37|61.60|
> ||quarot|11.70|36.18|58.46|34.40|69.53|59.91|51.70|
> ||our|9.65|41.13|66.25|39.40|74.21|61.72|56.54|
> ||quarot+nvfp4|9.90|39.85|64.31|38.00|73.50|64.64|56.06|
> ||ours+nvfp4|8.91|42.06|65.82|40.00|74.92|64.88|57.54|
> ||quarot 3bit|105.36|23.72|30.68|23.00|52.56|49.41|35.87|
> ||ours 3bit|16.04|30.12|48.82|29.60|64.20|56.27|45.80|
> |3.2 1B|baseline|9.75|36.26|60.47|37.20|74.53|60.61|53.81|
> ||quarot|19.75|26.71|45.33|30.40|62.57|52.17|43.44|
> ||our|13.58|30.89|50.63|31.80|69.59|54.54|47.49|
> ||quarot+nvfp4|13.88|31.57|51.01|31.60|68.06|56.27|47.70|
> ||ours+nvfp4|12.84|31.74|55.77|34.20|70.89|56.99|49.92|
> ||quarot 3bit|255.90|23.46|30.64|23.80|53.59|52.64|36.83|
> ||ours 3bit|27.54|25.09|39.73|27.80|60.45|52.17|41.05|
>
> The results show that for these more quantization-sensitive models, there remains considerable room for improvement over QuaRot. In the 3-bit setting, the benefits of rotation matrix optimization become even more pronounced. This supports our hypothesis that the more aggressive the compression, the more valuable such optimization becomes.
>
> Regarding NvFP4, although the Whip loss was originally designed for uniform quantization and may not be optimal for non-uniform formats like FP4, we find that the distribution tightening it induces still benefits the overall quantization. As a result, DartQuant + NvFP4 yields measurable gains over QuaRot + NvFP4.
>
> Looking ahead, we plan to explore alternative loss functions that are more tailored to the characteristics of FP4 and other non-uniform quantization formats, to further improve performance in such settings.
>
> In summary, we believe that for more difficult quantization scenarios—especially low-bit formats and challenging models—exploring rotation matrix optimization remains a promising and meaningful direction.

---

> > ### Comment · Reviewer_gzqM · 2025-08-05
> > **Official Comment**
> >
> > Thank you so much for providing the answers to my questions:
> >
> >
> > >For instance, in the 4-4-4 setting (Table 2), DartQuant improves the performance of LLaMA 3 8B from QuaRot’s 12.29/57.32 (PPL/Zero-Shot) to 10.78/62.38, and LLaMA 3 70B from 11.38/61.50 to 8.13/69.05. These are non-trivial improvements.
> >
> > I agree with the author that these are non-trivial improvements. However, the gap with the original model remains significant—for example, in the 4-4-4 case on LLaMA3-8B, there's a drop of around 2 perplexity points, which is substantial. In configurations where DartQuant performs closer to the baseline—such as 4-4-16 on LLaMA3-8B—it achieves 0.1 PPL points better than QuaRot and only 0.01 worse than SpinQuant. I also believe that the improvement could be more noticeable at lower precisions (e.g., 2-2-4), but the resulting accuracy drop is so large that it raises concerns about the practicality of the approach.
> >
> >
> > >We would also like to emphasize that, in the original paper, we prioritized extreme optimization efficiency by using a very small number of calibration samples. However, if one trades some speed for better accuracy, DartQuant benefits significantly from larger calibration sets (512 samples same as [1], SpinQuant use 800 samples). Detailed results are as follows:
> >
> > Thank you so much for providing more results. However, how many data points did you use for QuaRot in the above table? It would be great if we can see the results on the same number of calibration samples for DartQuant, QuaRot, and SpinQuant.
> >
> >
> > >Specifically, we implemented NvFP4 quantization using the FP4 codebase from the microsoft/microxcaling repository. Due to the use of complex secondary quantization in NvFP4, the inference speed is substantially reduced. Given limited time, we report results only on the two smaller models mentioned above. The quantitative results are presented below.
> >
> > Thank you so much for providing this nice table. After carefully looking at the table, I agree that for NvFP4, DartQuant outperformed QuaRot (by 1 PPL and 1.5% zero-shot accuracy) on 3B model.
> >
> > In the 3-bit case, as mentioned above, the gap between the baseline and the quantized models is large for both DartQuant and QuaRot schemes, and I believe the results above confirm this.
> >
> >
> > In conclusion, I will stick with my current score. I would like to thank the authors for their responses and efforts.

---

> ### Author Response · Authors · 2025-08-04
>
> Dear reviewer gzqM,
>
> We would like to thank you for your thoughtful comments and suggestions for our paper. We hope our response has addressed your questions. If you have any further concerns, we would be happy to clarify them. If our response has resolved your concerns, we would greatly appreciate your positive feedback and a possible improvement in your score.
>
> Thanks!

---

> ### Author Response · Authors · 2025-08-05
>
> Dear Reviewer gzqM,
>
> Thank you very much for your response! We would like to address your further concerns promptly and thoroughly one by one.
>
> >I agree with the author that these are non-trivial improvements. However, the gap with the original model remains significant—for example, in the 4-4-4 case on LLaMA3-8B, there's a drop of around 2 perplexity points, which is substantial. In configurations where DartQuant performs closer to the baseline—such as 4-4-16 on LLaMA3-8B—it achieves 0.1 PPL points better than QuaRot and only 0.01 worse than SpinQuant.
>
> First, we appreciate your recognition of the performance improvements achieved under the 4-4-4 setting.
>
> In response to your doubts about the marginal gains of our method over QuaRot at 4-4-16 on LLaMA3-8B, we wish to clarify that DartQuant reduces PPL relative to QuaRot by 1.16 (from 11.74 to 10.58). I guess you are talking about the results of the 4-8-16 setting. As a result, to further demonstrate the superiority of DartQuant over QuaRot, we additionally provide a comparison using 512 calibration samples (for details about the alignment of calibration set sizes, please refer to the clarification in point 2). The comparative results are shown in the table below:
>
> |Method|Model|WIKI|ARC_C|ARC_E|OBQA|PIQA|WG|AVG|
> |-|-|-|-|-|-|-|-|-|
> |baseline|3 8B|6.14|53.24|77.82|45.00|80.96|73.32|66.07|
> |quarot| |6.55|51.79|77.40|43.80|79.49|74.11|65.31|
> |Ours(paper)| |6.50|52.65|78.24|44.80|80.41|73.24|65.87|
> |Ours(512)| |6.31|52.98|78.13|44.80|80.65|73.28|65.96|
>
> As indicated, DartQuant reduces PPL loss by more than 50% compared to QuaRot. We would like to emphasize once again that, when a certain amount of optimization time is acceptable, DartQuant also delivers significant performance gains over QuaRot under the 4-8-16 setting.
>
> >Thank you so much for providing more results. However, how many data points did you use for QuaRot in the above table? It would be great if we can see the results on the same number of calibration samples for DartQuant, QuaRot, and SpinQuant.
>
> There are two places where calibration samples are used. One for rotation matrix optimization, and one for the GPTQ weight quantization process.
>
> For rotation matrix optimization: (1) QuaRot employs only random Hadamard matrices for rotation quantization, without optimizing the Hadamard matrices; thus, no data points involved, and the results presented in our table are directly from the original QuaRot experiments. (2) SpinQuant utilizes 800 data points to optimize the rotation matrices end-to-end. We use 128/512 samples for DartQuant, which is less than SpinQuant. Despite the smaller calibration set, as has shown in the rebuttal, DartQuant still significantly outperforms SpinQuant. In summary, the table provided in our rebuttal reflects a fair comparison. Furthermore, to strictly align with SpinQuant, we will report the results of DartQuant with 800 samples very soon within the discussion period.
>
> In addition, if your concerns pertain to the calibration set size used in the GPTQ quantization process, we clarify that, in all experiments (both in the original paper and the rebuttal), we used 128 calibration sets of length 2048 sampled from WikiText2, ensuring complete fairness in this regard as well.
>
> ***Further explainations about the value of DartQuant***
>
> It seems that our biggest disagreement lies in whether DartQuant has its significant contribution considering the accuracy drop for the 4-4-4 case. From your viewpoint, although DartQuant has a non-trivial improvement over QuaRot, there is still a drop of around 2 perplexity on models harder to quantize (like LLaMA3-8B), which makes DartQuant useless in practice. While we value your perspective, we respectfully hold a different opinion.
>
> First, for some circumstances where accuracy is not very sensitive, the 4-4-4 deployment is still a good choice considering the efficiency and accuracy trade-off.
>
> Second, although DartQuant still have more than 2 perplexity drop by now, we believe it already achieves a significant improvement over previous methods. This is significant, because as an researcher in this field, what we are doing is to push the quantization limit step by step, and we believe DartQuant is an meaningful step towards more efficient and accurate PTQ method for LLMs. Just like QuaRot and SpinQuant, which have not solved the W4A4KV4 quantization problem, but they have inspired the following research to make a further improvement. And we believe our findings in DartQuant paper could also inspire further research in this field. Thus, we sincerely request you to evaluate our contributions to this field.

---

> > ### Author Response · Authors · 2025-08-06
> >
> > 4-8-16
> > |Method|Model|WIKI|ARC_C|ARC_E|OBQA|PIQA|WG|AVG|
> > |-|-|-|-|-|-|-|-|-|
> > |baseline|3 8B|6.14|53.24|77.82|45.00|80.96|73.32|66.07|
> > |quarot| |6.55|51.79|77.40|43.80|79.49|74.11|65.31|
> > |spinquant(800)| |6.55|51.88|77.10|44.80|79.92|74.19|65.58|
> > |Ours(paper)| |6.50|52.65|78.24|44.80|80.41|73.24|65.87|
> > |Ours(512)||6.31|52.98|78.13|44.80|80.65|73.28|65.96|
> > |Ours(800)||6.30|52.88|77.61|45.00|80.09|74.43|66.00|
> >
> > We appreciate the reviewer’s suggestion to include fair comparisons. As shown in the table above, SpinQuant is evaluated with 800 calibration samples. Under the same setting (800 samples), our method achieves clearly better average accuracy (our 66.00 vs. spinquant 65.58) and PPL(our 6.30 vs. spinquant 6.55) . Even with only 512 calibration samples, our method still outperforms SpinQuant (our 65.96 vs. spinquant 65.58), demonstrating its efficiency in low-data regimes.
> >
> > Beyond accuracy, DartQuant brings significant practical advantages: it achieves up to 47× faster optimization, reduces memory usage by nearly 10× during the optimization phase for Llama 3 70B, and requires no additional hyperparameter tuning. Its plug-and-play nature, combined with consistent improvements in both speed and accuracy, makes DartQuant highly practical and scalable—especially important as models and deployment demands continue to grow.
> >
> > We hope this response addresses your concerns, and we would greatly appreciate your feedback.

---

> > > ### Comment · Reviewer_gzqM · 2025-08-06
> > > **Official Comment**
> > >
> > > Thank you for providing additional data and comparing it against SpinQuant using the same number of samples. Now, I am convinced that DartQuant can improve the accuracy and will raise my score.

---

> > > > ### Author Response · Authors · 2025-08-06
> > > >
> > > > Thank you for your thoughtful consideration and positive feedback. We sincerely appreciate your recognition and support.

---

### Official Review · Reviewer_SzJU · 2025-06-28

**Clarity:** 4
**Significance:** 3
**Originality:** 3
**Rating:** 5
**Confidence:** 3

**Summary:**

This paper propose DartQuant, a post-training weight-activation-kv cache quantization method for LLMs, that propose a new layer-wise loss to learn the rotation matrices to flatten the activation distribution without end-to-end fine-tuning. This method reduces the quantization time (computational complexity) but achieves SoTA model performance.

**Questions:**

1. Could the author clarify the quantization format for activations? I assume the weight matrix is quantized using groupwise scales and integer quantization. What about activations?
2. The Whip loss is impressive. Does the Whip loss assume the activation distribution is symmetric over zero? What if the activation mean is not zero?
3. In Algorithm 1, what does the function "Row_sampling(X)" mean?
4. Could the author give more information about the objective in Algorithm 1 like variance across $Z$ initialized with different random seeds and learning rates? Is the problem easy to solve for gradient descent?

**Ethical Concerns:**

["NO or VERY MINOR ethics concerns only"]

**Final Justification:**

Thanks the author for clarification. All of my questions have been answered. I remain positive with this work.

**Limitations:**

As stated in the limitation section of the paper, DartQuan did not explore the number formats that may have native hardware support like FP4. INT4 activation is not natively supported by most of GPUs. Other concerns are reflected by my questions.

**Quality:**

3

**Strengths And Weaknesses:**

## Strengths
- A good summary of related work on rotation-based PTQ methods for LLMs. This serves as a good background intro for readers who are not very familiar with this area.
- A clear description of the method supported by sufficient explainations. The whip loss design is simple but effective.
- Extensive experiments and ablation study to demostrate the effectiveness of the method

## Weaknesses
I did not see partitical weaknesses though I have a few questions about the details (See Question Section).

If will be helpful if the author show some inference latency/throughput/memory results even though it may be the same as other ratation-based methods like SpinQuant.

---

> ### Author Rebuttal · Authors · 2025-07-30
>
> We sincerely thank the reviewer for the highly positive feedback and your thoughtful questions. We deeply appreciate your recognition of our work. We provide our point-by-point responses to the raised questions below.
>
> **Weakness:** *If will be helpful if the author show some inference latency/throughput/memory results even though it may be the same as other ratation-based methods like SpinQuant.*
>
>  As suggested, we evaluated the inference performance of DartQuant (W4A4KV4) using LLaMA 2-7B on single A800 GPU under various batch sizes and input lengths using randomly sampled prompts and compared it with FP16 baseline. Specifically, we report prefill time, decoding speed (fixed 50-token generation), and peak memory usage (torch.cuda.max_memory_allocated()). Results are provided below:
>
> |Batch|SeqLen|Prefill(ms)DartQuant|Prefill(ms)FP16|Decode(tok/s)DartQuant|Decode(tok/s)FP16|Mem(GB)DartQuant|Mem(GB)FP16|
> |--|--|--|--|--|--|--|--|
> |1|2048|209|281|20.43|14.08|3.89|13.66|
> |4|2048|611|907|81.75|52.71|4.72|16.79|
> |8|2048|1138|1452|163.7|104.32|5.81|20.95|
> |1|1024|108|141|20.5|12.99|3.77|13.16|
> |4|1024|306|482|81.87|47.29|4.19|14.75|
> |8|1024|567|752|167|109.32|4.74|16.89|
> |1|512|61|83|21.28|12.43|3.70|12.9|
> |4|512|163|257|82.91|46.55|3.92|13.74|
> |8|512|288|395|168.35|112.33|4.21|14.85|
>
> **Question1:** *Could the author clarify the quantization format for activations? I assume the weight matrix is quantized using groupwise scales and integer quantization. What about activations?*
>
> Thank you for raising this. As stated in Line 258,  all activations are quantized using per-token asymmetric  integer quantization.
>
> **Question2:** *The Whip loss is impressive. Does the Whip loss assume the activation distribution is symmetric over zero? What if the activation mean is not zero?*
>
> This is an excellent question that highlights a potential limitation of the current Whip design. Indeed, Whip implicitly assumes that the activation distribution is approximately zero-mean. While this assumption often holds in practice, it is not strictly guaranteed.
>
> When the activation mean deviates significantly from zero, the effectiveness of Whip may be reduced. In such cases, applying zero-mean normalization to the activations before computing the loss could yield more stable optimization. We will add a discussion of this assumption and its implications in the revised version. We sincerely thank the reviewer for this insightful observation.
>
> **Question3:** *In Algorithm 1, what does the function "Row_sampling(X)" mean?*
>
> "Row_sampling(X)" refers to randomly sampling a subset of activation tokens from the full matrix for gradient computation in each optimization step. This strategy improves efficiency by reducing computation without sacrificing performance.
>
> To avoid potential confusion caused by the name, we will rename this function to "Token_sampling(X)" in the revised version for better clarity.
>
> **Question4:** *Could the author give more information about the objective in Algorithm 1 like variance across
>  initialized with different random seeds and learning rates? Is the problem easy to solve for gradient descent?*
>
> Thank you for this thoughtful question. As the optimization involves random sampling, the process is inherently non-deterministic and not exactly reproducible across different random seeds.
>
> Nonetheless, we conducted multiple independent runs under varying random conditions (implicitly covering different seeds and sampling paths). As shown below, the final loss consistently converges to similar values across runs, and with comparable iteration counts. This empirical evidence suggests that the optimization problem in **Algorithm 1** is well-conditioned and tractable via gradient descent, without suffering from instability or high variance.
>
> |ID|LR|FinalObjective|Iterations|
> |-|-|-|-|
> |1|1.00E-03|2004.9094|138|
> |2|1.00E-03|2004.9535|141|
> |3|1.00E-03|2004.9191|140|
> |4|2.00E-03|2004.9507|131|
> |5|5.00E-03|2004.8560|123|

---

> > ### Comment · Reviewer_SzJU · 2025-08-03
> >
> > Thanks the author for clarification. I remain positive with this work.

---

> > > ### Author Response · Authors · 2025-08-04
> > >
> > > Dear reviewer SzJU:
> > >
> > > Thank you for your positive and constructive feedback. We are grateful that we have addressed your concerns.

---

### Official Review · Reviewer_sxB6 · 2025-07-02

**Clarity:** 3
**Significance:** 3
**Originality:** 2
**Rating:** 4
**Confidence:** 5

**Summary:**

This paper introduces DartQuant, a post-training quantization method for large language models. The method aims to significantly reduce the computational cost associated with calibrating rotation matrices. DartQuant builds upon the computationally invariant framework and optimizes rotation matrices to improve uniform quantization. It consists of two key components:
1. Whip Loss – a loss function designed to reshape activation distributions toward uniformity.
2. QR-Orth – an optimization scheme using QR decomposition to enforce orthogonality of the rotation matrices.

The authors report a 47× speedup and 10× memory reduction, enabling the calibration of a 70B model on a single consumer GPU.

**Questions:**

1. Can you explain the better convergence rate of QR-orth method compare to Caley optimization?
2. Can you compare to [5] which should be easily doable within the rebuttal period, and [3,4] if you have additional time? (See strengths  and weaknesses)
3. Can you add more models (Mistral, Qwen, or Phi) and mathematical reasoning tasks such as GSM8K, MathQA?

**Ethical Concerns:**

["NO or VERY MINOR ethics concerns only"]

**Final Justification:**

The authors addressed all of my concerns, hence I'm increasing my score.

**Limitations:**

The only limitation mentioned is "tailored for uniformly distributed integer formats". Is there no other limitations?

**Quality:**

2

**Strengths And Weaknesses:**

The paper addresses a highly practical and relevant challenge: reducing the cost of optimizing rotation matrices during LLM quantization.
Compared to prior methods such as SpinQuant, DartQuant offers substantial speed and memory benefits, making it more accessible for users with limited hardware. The paper provide a strong result on extreme qunatization scenario like 4 bit qunatization.

One of the biggest weakness is that all experiments are conducted exclusively on LLaMA models. The generality of DartQuant should be evaluated on other architectures like Mistral, Qwen, or Phi to demonstrate broader applicability. Moreover, adding experiments on mathematical reasoning tasks such as GSM8K, MathQA or longbench would strengthen the empirical evaluation and show the method’s performance on more challenging domains.

Another big weakness is the missing comparisons with closely related quantization methods, including some mixed-precision quantization method, e.g. at least [3, 4].

Another big weakness is the unsubstantiated dismissal of kurtosis-based methods. Reference [5] demonstrates the effectiveness of kurtosis as a quantization signal. Dismissing it without a rigorous comparison weakens the argument for Whip Loss. A head-to-head evaluation and deeper justification for the superiority of Whip Loss are needed. This comparison is key since with KurTail [5] we can also train the rotation matrices single consumer GPU similar to DartQuant.

The justification for the “Whip Loss” function is vague. It is said to be “inspired by” transforming Laplace to uniform distributions, yet the specific formulation, ∑exp(−|xᵢ|), is heuristic and lacks theoretical derivation. The paper does not explore alternative formulations or explain why this one is particularly effective at promoting uniformity.

The paper reports a 1.4× per-iteration speedup (Table 4) but also mentions a 41× total speedup (line 375) and a 47× training speedup (line 332). I suggest the authors to provide a section and ablation study to clarify and justify each speed up.

The QR-Orth optimization appears more effective than methods such as Cayley optimization, which contradicts [1]. The paper does not justify its choice or acknowledge this performance gap. The slower convergence rate in Caley methods may be due to suboptimal hyperparameter selection. Moreover, QR-Orth is not a novel method. The QR-Orth scheme is well-known in manifold optimization. The current presentation can be potentially misleading and be read as a novel contribution. The authors should cite foundational works,
such as Optimization Algorithms on Matrix Manifolds [2].

Minor comment: The exact optimization process for rotation matrices is unspecified. The details of how the rotation matrices are optimized are under-explained. For instance, the specific optimization process for Rotations is not described. Further the paper just claimed that they can optimise llama 70B on 1 GPU but running a llama 70B in fp16 needs 120GB for inference. Do they use a larger gpu to save the activation or they used methods like offloading to run their experiment clarifying the optimization process would help paper clarity.

Figure 11 presents the distribution before rotation is applied. Including the post-rotation distribution would help demonstrate the effectiveness of DartQuant more convincingly.

References:
1. Efficient Riemannian Optimization on the Stiefel Manifold via the Cayley
Transform, ICLR 2020
2. Optimization Algorithms on Matrix Manifolds, Princeton University Press, 2009.
3. ResQ: Mixed-Precision Quantization of Large Language Models with Low-Rank
Residuals, ICML 2025
4. DuQuant: Distributing Outliers via Dual Transformation Makes Stronger
Quantized LLMs (Neurips 2024)
5. KurTail: Kurtosis-based LLM Quantization, SLLM workshop ICLR 2025

---

> ### Author Rebuttal · Authors · 2025-07-30
>
> We thank the reviewer for their valuable comments and suggestions. We have thoroughly reviewed each point and provide our detailed responses below.
>
> **Weakness1&Qusetion3** *More model & task*
>
> In **Appendix H**, we experimentally demonstrate the effectiveness of our method on two MoE models (Mistral-8x7B and Deepseek-moe-16B). Following your suggestion, we also conducted comparative experiments on Qwen, Phi, and Mistral models across 5 zero-shot tasks, as well as additional evaluations on math reasoning datasets GSM8K and MathQA.
>
> - Our method significantly outperforms QuaRot in perplexity, zero-shot accuracy, and math reasoning on all three models.
>
> - Notably, on the math reasoning task GSM8K, our method achieves a remarkable improvement over QuaRot (e.g., a 16.52% increase on Qwen 2.5 3B), indicating promising advantages on tasks sensitive to quantization complexity.
>
> |Method|Model|WIKI|ARC_C|ARC_C|OBQA|PIQA|WG|5 AVG|GSM8K|MQA|MMLU|
> |-|-|-|-|-|-|-|-|-|-|-|-|
> |baseline|mistral 7B|5.25|53.92|79.75|43.80|81.99|74.03|66.70|37.75|35.94|58.71|
> |quarot||5.86|45.73|76.60|41.40|77.64|70.80|62.43|26.84|30.95|50.72|
> |Ours||5.57|49.91|77.81|42.00|80.68|71.58|64.40|30.85|33.46|53.96|
> |baseline|qwen2.5 3B|8.03|47.44|73.31|42.60|78.94|68.19|62.10|75.89|37.28|65.09|
> |quarot||9.85|43.51|67.55|38.80|74.04|62.74|57.33|40.56|34.17|55.51|
> |Ours||9.03|43.17|71.38|40.20|75.46|63.93|58.83|57.08|36.31|58.98|
> |baseline|phi 3 mini|6.02|56.65|78.57|47.80|80.90|73.95|67.57|80.21|39.59|70.15|
> |quarot||8.09|46.50|73.40|41.80|74.86|63.72|60.06|50.34|34.07|57.45|
> |Ours||7.64|49.23|75.17|44.40|76.93|67.71|62.69|59.21|34.50|59.94|
>
> Due to cost, we only ran LongBench on LLaMA 3 8B, where our method nearly halves the performance degradation compared to QuaRot. Different subsets of Longbench use different metrics (including F1 score, Rouge, etc.). For simplify, we used the average of these scores to evaluate the overall performance. The comparison of scores for each subset can also be seen in the following table.
>
> |Method|**longbench avg**|2wikimqa|hotpotqa|musique|multifieldqa_en|multifieldqa_zh|narrativeqa|qasper|triviaqa|gov_report|qmsum|
> |-|-|-|-|-|-|-|-|-|-|-|-|
> |baseline|**27.91**|11.50|9.55|6.12|20.55|21.20|5.04|18.56|90.78|27.50|23.90|
> |quarot|**23.52**|10.44|8.10|5.00|20.17|15.50|5.17|18.66|81.90|19.03|20.27|
> |ours|**25.61**|11.80|9.47|5.86|23.54|19.46|6.21|19.30|87.47|21.30|21.93|
>
> |Method|vcsum|dureader|lcc|repobench_p|passage_retrieval_en|passage_retrieval_zh|passage_count|trec|lsht|multi_news|samsum|
> |-|-|-|-|-|-|-|-|-|-|-|-|
> |baseline|6.67|13.16|71.98|68.40|9.17|8.02|2.00|75.00|34.50|17.65|44.84|
> |quarot|4.18|13.88|58.25|57.44|3.31|5.17|2.55|65.00|26.75|12.41|40.74|
> |ours|4.85|13.34|63.68|60.00|4.24|6.88|3.17|67.00|28.25|15.79|44.33|
>
> **Weakness2&Qusetion2** *Mixprecision comparison*
>
> In **Appendix F**, we provide detailed comparisons with two classical mixed-precision quantization methods (QUIK and Atom). In line with your advice, we also compare with [3] and [4]. Using the publicly available code from [3,4], after aligning quantization settings, we performed the comparisons. Note that [3] does not support Mistral, and [4] doesn't support Qwen, so these results are omitted. The results are as follows:
>
> |Method|Model|WIKI|ARC_C|ARC_C|OBQA|PIQA|WG|AVG|
> |-|-|-|-|-|-|-|-|-|
> |baseline|qwen2.5-3B|8.03|47.44|73.31|42.60|78.94|68.19|62.10|
> |[3]||8.89|42.75|68.77|40.80|76.77|64.64|58.75|
> |Ours||9.03|43.17|71.38|40.20|75.46|63.93|58.83|
> |baseline|3 8B|6.14|53.24|77.82|45.00|80.96|73.32|66.07|
> |[3]||7.08|49.49|74.79|42.80|78.67|69.14|62.98|
> |[4]||8.13|44.62|72.14|40.80|75.52|68.03|60.22|
> |Ours||7.43|48.63|75.52|43.00|78.18|70.68|63.20|
> |baseline|mistral-7B|5.25|53.92|79.75|43.80|81.99|74.03|66.70|
> |[4]||5.78|45.82|75.33|42.20|79.43|69.93|62.54|
> |Ours||5.57|49.91|77.81|42.00|80.68|71.58|64.40|
>
> From the result, DartQuant performs comparably to [3] and better than [4]. Notice that, [3] achieves higher accuracy by combining rotation with high-bit outlier protection, while DartQuant improves low-bit quantization precision through optimizing the rotation matrix. The two methods are algorithmically orthogonal and can be combined for further improvements.
>
> **Weakness3** *vs [5]*
>
> We have reported Whip vs. kurtosis results under DartQuant settings in **Appendix I**, demonstrating Whip’s superiority. To further validate this, we compared Whip to the method in [5]. As [5]’s code is unavailable and the authors have not yet responded to our inquiries, we re-implemented their approach strictly following their published settings, using 512 calibration samples (128 samples in paper for efficiency). The following results (with [5]’s paper results as reference) show that, with equal calibration samples, our Whip loss achieves significantly better performance on two different models. This aligns with Figures 6 (lines 395-362) and Appendix I, confirming that Whip pushes activations closer to uniformity, outperforming other losses.
>
> |Method|Model|WIKI|ARC_C|ARC_C|OBQA|PIQA|WG|AVG|
> |-|-|-|-|-|-|-|-|-|
> |baseline|2 7B|5.47|46.33|74.54|44.20|79.05|69.06|62.64|
> |[5]||5.90|43.10|72.00|41.20|76.60|66.80|59.94|
> |Ours||5.72|45.09|72.72|43.40|78.18|68.98|61.67|
> |baseline|3 8B|6.14|53.24|77.82|45.00|80.96|73.32|66.07|
> |[5]||7.20|48.20|75.40|43.60|78.40|70.00|63.12|
> |Ours||6.84|51.02|76.47|44.00|79.27|71.67|64.49|
>
> **Weakness4** *Why whip*
>
> The Whip loss formulation follows the transformation in Eq. 3. As shown in Fig. 5, Eq. 3 expands the near-zero region of the original distribution to a larger range. The Whip gradient near zero is much larger than that at the tails (due to the exponential function), which is key to Whip’s efficiency, as it rapidly pushes small values up in the rotated data. Under norm-invariant constraints, outliers are quickly suppressed.
>
> We also use Llama 3 8B tested two functions with similar shapes but gentler gradients than Whip:
>
> - $L1= \sum ReLU(1-|x_i|/\alpha)$, where $\alpha$ is set to three times the median (outlier threshold)
>
> - $L2= \sum 1/(|x_i|+1)$
>
> |Loss|WIKI|ARC_C|ARC_C|OBQA|PIQA|WG|AVG|
> |-|-|-|-|-|-|-|-|
> |baseline|6.14|53.24|77.82|45.00|80.96|73.32|66.07|
> |quarot|8.40|40.02|66.33|41.00|73.61|66.77|57.55|
> |L1|7.56|48.48|72.93|42.80|77.58|68.03|61.96|
> |L2|7.65|46.25|72.85|41.20|76.39|70.01|61.34|
> |whip|7.43|48.63|75.52|43.00|78.18|70.68|63.20|
>
> With identical hyperparameters, these alternative losses show inferior performance compared to Whip, demonstrating Whip’s advantage.
>
> **Weakness5** *Speedup detail*
>
> We clarify that QR-Orth achieves a 1.4× per-iteration speedup over Cayley (Table 4), a 41× “total speedup” to converge to the same loss (line 375), and a 47× speedup in rotation matrix training compared to SpinQuant (line 332).
>
> **Weakness6&Qusetion1** *QR-Orth vs Cayley*
>
> Section 5 of [1] points out that the Cayley transform’s iterative form is a contraction mapping, with Theorem 1 proving convergence only if step size $\alpha < 2 / \|W\|$. Otherwise, updates may diverge or leave the manifold, limiting Cayley SGD’s step size and slowing convergence.
>
> QR-Orth, however, projects updated parameters back onto the manifold without this constraint, allowing larger update steps and faster convergence.
>
> |optim|Delta|
> |-|-|
> |QR-Orth|1.20E-04|
> |Cayley|8.95E-05|
>
> In the same hyperparameter settings, we measured the average parameter update magnitude over 10 iterations: QR-Orth’s updates are 1.2× larger than Cayley’s, consistent with the reasoning above.
>
> Notably, the Cayley implementation follows SpinQuant’s original hyperparameter configuration in paper, which has been tuned and validated for this task. To rule out suboptimal hyperparameter effects, we tested multiple learning rates. Across all tested settings, QR-Orth achieved consistently lower losses than Cayley, confirming our claims.
>
> |Optim|LR|Loss|Optim|LR|Loss|
> |-|-|-|-|-|-|
> |Cayley|0.01|1577.49|QR-Orth|0.01|1530.68|
> ||0.1|1565.52||0.1|1537.05|
> ||1|1565.71||1|1527.65|
> ||2|1566.04||2|1526.59|
> ||5|1565.53||5|1531.07|
>
> Finally, we sincerely thank the reviewer for pointing out the lack of citation. We acknowledge that QR-based orthogonalization is a well-established tool in manifold optimization [2], and we do not claim novelty in the algorithm itself. Our contribution lies in being the first to adapt QR-Orth for rotation-matrix training in LLM post-training quantization. We will revise the final version to clarify this point and avoid any overstatement.
>
> **Weakness7** *Memory usage*
>
> As Fig. 4 illustrates, DartQuant’s process has two stages. First, the LLM runs once on the calibration set to collect activations stored locally. Then rotation matrices are trained offline on this stored data, so no large model loading is needed during rotation optimization. For large models such as LLaMA 70B, we adopt a layer-wise loading strategy allowing inference to be performed on a single GPU. We further clarify these details in **Appendix K** to avoid misunderstandings.
>
> Finally, we plan to release our code to facilitate reproducibility within the community.
>
> **Weakness8** *Visualization*
>
> Thank you for your suggestion. We will add histograms of post-rotation activation distributions in the revision to better demonstrate DartQuant’s effectiveness.
>
> **Limitations**
>
> In addition to being tailored for uniform quantization formats, we acknowledge a potential limitation regarding the distributional assumption in the Whip loss. Specifically, Whip assumes that activations are approximately zero-mean, which generally holds true for most transformer layers. However, in rare cases where the activation mean significantly deviates from zero, the effectiveness of Whip may degrade. As a remedy, we suggest applying mean-centering or zero-mean normalization prior to optimization, and we will explicitly include this consideration in the revised version. We also plan to explore adaptive variants of Whip that are robust to non-zero-mean activations in future work.

---

> > ### Author Response · Authors · 2025-08-04
> >
> > Dear Reviewer sxB6,
> >
> > We sincerely thank you for your thoughtful comments and constructive suggestions on our paper. We hope that our responses have adequately addressed your concerns. Should you have any further questions or require additional clarification, we would be glad to assist. If our responses have resolved your concerns, we would greatly appreciate your positive feedback and a potential improvement in your score.
> >
> > Thanks!

---

> > > ### Author Response · Authors · 2025-08-06
> > >
> > > Dear Reviewer sxB6,
> > >
> > > We are writing to kindly follow up on our rebuttal submission, and to check whether you might have any further feedback or comments.
> > >
> > > We greatly appreciate the time and effort you have invested in evaluating our work, and your input is very valuable to us. If there’s anything we can clarify or provide further, we’d be happy to do so.
> > >
> > > Thank you again for your time and consideration.

---

> > > > ### Comment · Reviewer_sxB6 · 2025-08-06
> > > >
> > > > Thank you for the detailed and the additional experimental results.
> > > >
> > > > Can you please clarify the quantization scheme used for the new results that you reported in the rebuttal? Some of the reported results seem (significantly) better than what was reported in the original papers which is somewhat surprising. For example your results for ResQ are better than the results in their paper. Is the experimental setting equivalent or are there important differences that contribute?

---

> > > > > ### Comment · Reviewer_sxB6 · 2025-08-06
> > > > >
> > > > > Can you also please explain between the discrepancy between the reported results in your appendix Table 18 and the results here in the rebuttal? For example, Table 18 the results for DartQuant on WIKI are: 5.90 for 2-7b but in the rebuttal you state 5.72. Similarly, for 3-8b you report 7.29 in Table 18 but your report 6.84 in one of the tables in the rebuttal and 7.43 in the other table.

---

> > > > > > ### Author Response · Authors · 2025-08-07
> > > > > >
> > > > > > We sincerely thank the reviewer for your feedback. Your attention to detail and your further questions are appreciated.
> > > > > >
> > > > > > We apologize for not clearly detailing our experimental settings in the initial rebuttal due to the word limit. Below, we provide a more comprehensive explanation:
> > > > > >
> > > > > > First, we would like to further clarify the optimization procedure of DartQuant, which is also outlined in Algorithm 1 of our paper (on page 6). To calibrate the rotation matrix, we used 128 calibration sequences of length 2048 (i.e., 2048 tokens for each sequence) sampled from WikiText2. For each layer, we first input the calibration set into the LLMs to extract the input features, which are saved on local. Then we use these extracted features to calibrate the rotation matrices with Whip loss using SGD. For simplicity, in the SGD process, we didn’t use all of the extracted features, but only randomly sample 10% of the token features (Line 4 of Algorithm 1).
> > > > > >
> > > > > > >Can you please clarify the quantization scheme used for the new results that you reported in the rebuttal? Some of the reported results seem (significantly) better than what was reported in the original papers which is somewhat surprising. For example your results for ResQ are better than the results in their paper. Is the experimental setting equivalent or are there important differences that contribute?
> > > > > >
> > > > > > In the table of “answers to Weakness1&Qusetion3 More model & task” part, we use W4A4KV16 for all experiments. During the extended model evaluations, we found that the Phi model uses an attention head dimension of 96, which is not a power of 2. As a result, it is incompatible with the fast Hadamard transform, preventing us from applying online Hadamard-based quantization to the KV matrices. Due to time constraints and our intention to follow your suggestions by including comparisons with all referenced models and methods, we use 4-4-16 quantization setting. Thus, ***for models of Mistral 7B, Qwen2.5 3B and phi 3 mini, we use the W4A4KV16 throughout the rebuttal*** for consistency.
> > > > > >
> > > > > > To make the results reliable, we reproduce the results of ResQ using the code and settings from its official repository, except for using FP16 for KV cache. As a result, our reproduced ResQ (4-4-16) results on Qwen2.5-3B are better than those reported in the original paper, which is 4-4-4 quantization. For example, on Qwen2.5-3B, ResQ reported a PPL of 9.0 using 4-4-4, while our reproduced result using 4-4-16 is 8.89.
> > > > > >
> > > > > > >Can you also please explain between the discrepancy between the reported results in your appendix Table 18 and the results here in the rebuttal?
> > > > > >
> > > > > > In the table of “answers to Weakness3 vs [5]” part, we use W4A4KV4 for all experiments. The quantization settings used for the rebuttal is as follows:
> > > > > > - We follow the settings with [5] to use 512 calibration sequences of length 2048 sampled from WikiText2.
> > > > > > - To ensure a fair comparison, we also remove the token feature sampling operation (X←Row_sampling(X)) in Algorithm 1, which is also aligned with [5].
> > > > > > - We use QR-Orth SGD optimizer for 30 epochs, while [5] use Cayley Adam for 100 epochs.
> > > > > >
> > > > > > All other setting are the same with the paper. Under this setting, we can improve the PPL from 5.90->5.72 and 7.43->6.84 for LLaMA 2 7B and LLaMA 3 8B, respectively. In summary, the substantial accuracy improvement in the rebuttal results is primarily due to removing the sampling strategy and allowing for more sufficient optimization. While the settings in our original paper were designed for maximal efficiency, utilizing the full calibration set with sufficient optimization allows DartQuant to better optimize the rotation matrices and achieve its full performance potential.
> > > > > >
> > > > > > As for the 7.29 PPL of LLaMA 3 8B in Table 18, as shown in the paper, it is the result of the W4A4KV16 setting, corresponding to line 611 in Appendix F.
> > > > > >
> > > > > > We sincerely hope for your understanding that, due to the tight rebuttal schedule and strict word limit, we were unable to perform complete comparisons under all quantization settings. We hope this additional clarification can address your concerns. If you have any further questions regarding the experimental setups during the rebuttal period, please do not hesitate to let us know—we would be more than happy to clarify them promptly.

---

> > > > > > > ### Author Response · Authors · 2025-08-08
> > > > > > >
> > > > > > > Dear Reviewer,
> > > > > > >
> > > > > > > There are fewer than 20 hours remaining in the rebuttal discussion phase. We sincerely wanted to check whether our previous responses have addressed your concerns. Please let us know if there are any remaining questions, we would be happy to clarify them promptly.
> > > > > > >
> > > > > > > Thank you again for your time and consideration.

---

### Author Response · Authors · 2025-08-09
**Rebuttal Summary and Appreciation**

Dear Reviewers, ACs, SACs, and PCs,

We sincerely thank you for your time, effort, and constructive feedback throughout the review and rebuttal process. Your insightful comments have greatly improved the quality of our work, and we truly enjoyed the in-depth academic discussions with all reviewers.

**About DartQuant**

Through discussions with reviewers, we summarize the advantages of our work as follows: Our work introduces an **efficient rotation optimization framework** for low-bit quantization by integrating **Whip Loss** and **QR-Orth**, achieving a balance of simplicity and high performance.

**1)Extensive validation:** Demonstrated effectiveness on a variety of models, including LLaMA, Mixtral 7×8B, DeepSeek-MoE-16B (in the paper), and Qwen, Phi, Mistral (in the rebuttal).

**2)High performance:** Consistently strong results across diverse architectures and tasks, supported by comprehensive visualizations and ablation studies that validate the impact of Whip Loss and QR-Orth.

**3)Key contributions:** Provides up to **47× faster** rotation optimization and **10× reduction in peak memory** usage for 70B-scale models compared to state-of-the-art methods like SpinQuant and OSTQuant. Without considering cost constraints, DartQuant is capable of substantially enhancing accuracy on downstream tasks.

**Rebuttal highlights**

Through the rebuttal process, we addressed most concerns:

- Reviewer gzqM: “Now, I am convinced that DartQuant can improve the accuracy and will raise my score.”
- Reviewer SzJU: “Thanks the author for clarification. I remain positive with this work.”

Our efficiency and novelty were widely appreciated:

- Reviewer sxB6: “DartQuant offers substantial speed and memory benefits, making it more accessible for users with limited hardware.”
- Reviewer SzJU: “The Whip Loss design is simple but effective.”
- Reviewer gzqM: “The paper is well-written and easy to follow… The idea is interesting.”
- Reviewer uEAv: “DartQuant is a method for efficient rotational calibration in LLM quantization.”

We particularly valued the detailed exchanges with reviewer sxB6, which allowed us to present the effectiveness and scalability of DartQuant from perspectives beyond those originally covered in the paper. While the time limit prevented us from confirming that our final responses fully addressed all of his/her concerns, we sincerely hope they did.

Regarding reviewer uEAv’s remaining concerns on efficiency, we provided detailed clarifications and respectfully invite all reviewers and the AC to read that discussion for a deeper understanding of DartQuant’s advantages.

**Closing**

Once again, we thank all reviewers for their thoughtful input. We believe DartQuant offers a new perspective and a highly efficient solution to rotation matrix optimization in low-bit quantization, and can make a valuable contribution to NeurIPS 2025. We hope our work will receive your further support, while fully respecting your final decision.

Best regards,

All authors

---

### Decision · Program_Chairs · 2025-09-17

**Decision:**

Accept (poster)

**Comment:**

This paper presents DartQuant, an efficient method for calibrating rotation matrices in large language model post-training quantization. The method introduces a "Whip Loss" to reshape activation distributions for better quantization and a "QR-Orth" scheme for efficient optimization. The paper’s key strength, lauded by reviewers sxB6 and SzJU, is its massive improvement in efficiency, i.e., up to 47x speedup and 10x memory reduction, which makes rotation-based quantization for 70B models accessible on consumer hardware. Initial weaknesses, noted by sxB6 and gzqM, included limited experimental scope and marginal accuracy gains in some settings. However, the authors provided a comprehensive rebuttal with extensive new experiments across more models (Qwen, Phi) and tasks (GSM8K), convincingly demonstrating strong performance and scalability. This effort successfully swayed reviewers gzqM and sxB6 to support the paper. Although reviewer uEAv remained concerned about the focus on compression time, the consensus is that the dramatic reduction in computational cost is a significant and practical contribution. Therefore, I recommend acceptance.